



**StageIV-IRC – A High-resolution Dataset of Extreme Orographic**
**Quantitative Precipitation Estimates (QPE) Constrained to Water Budget**
**Closure for Historical Floods in the Appalachian Mountains**
Mochi Liao[1] and Ana P. Barros[1]
1. Civil and Environmental Engineering, University of Illinois Urbana-
Champaign, Urbana, IL
**Corresponding Author:**
Dr. Ana Barros
E-mail: barros@illinois.edu
Phone: +1 217-333-8038



## Abstract

Quantitative Flood Estimation (QFE) in complex terrain remains a grand challenge in operational hydrology due to the lack of accurate high-resolution Quantitative Precipitation Estimates (QPE) at spatial and temporal resolutions needed to capture the variability of orographic precipitation and where radar-based QPE are available there are significant biases due to the geometry and constraints of radar operations. Here, we present a high-resolution (i.e. 250m, 5minute-hourly) QPE dataset for 215 extreme (flood-producing) events from 2008 to 2024 for 26 gauged basins in the Appalachian mountains constrained to meet basin-scale water budget closure through inverse rainfall-runoff modeling to correct the Next Generation Weather Radar (NEXRAD) Stage IV analysis (4 km resolution, hourly) using a fully-distributed uncalibrated hydrological model that leverages recent advances in hydrologic modeling in mountainous regions (e.g. improved river routing and initial soil moisture estimation). The corrected Stage IV analysis QPE is referred to as StageIV-IRC (Inverse Rainfall Correction). Previously, a subset of this dataset informed the construction of a generalized QPE error prediction model and providing physics insights into orographic QPE uncertainties for various radar-based QPE products in complex terrain. The unique advantage of the StageIV-IRC QPE is it is in agreement with ground-based rainfall measurements and achieves water budget closure at the storm-flood event scale within observational uncertainty of streamflow observations when it is used to drive hydrological simulations of historical floods, that is the golden standard in hydrological modeling. The QPE dataset is publicly available at: https://doi.org/10.5281/zenodo.14028867 (Liao and Barros, 2024).



## 1. Introduction

Over the past few decades, extreme precipitation has become an increasingly important research topic due to its social, economic, and environmental impacts (Alimonti et al., 2022; Wernberg et al., 2013). Studies show that both total annual precipitation and extreme precipitation events have increased in the US and in other parts of the world during the last century (e.g. Milly et al., 2002), often resulting in floods (e.g. Pielke and Downtown, 2002), and flash floods in the context of complex terrain due to steep slopes (e.g. Schumacher, 2017; Czigány et al., 2010). Flash floods are characterized by fast rainfall runoff responses on the scale of a few hours (< 6 hours) after extreme precipitation events, with watershed areas often ranging from a few tens to hundreds of square kilometers (Borga et al., 2010; Lumbroso and Gaume, 2012). As one of the deadliest natural hazards, flash floods often are associated with landslide events (e.g. Gupta et al., 2016; Deijns et al., 2022) and cause severe life loss and property damage (Saharia et al., 2017; Špitalar et al., 2014) such as recently in the USA and in Spain. An estimated 94 million people are affected worldwide every year since 2000 (Guha-Sapir et al., 2018; Wu et al., 2020) and the average annual economic loss in the U.S. subjected to freshwater flooding exceeds USD $8 billion (Wing et al., 2018). A recent noteworthy example is Hurricane Helene in late September 2024, with a death toll over 200 in the Southeast U.S. Despite extensive studies on improving flash flood simulations in small headwater basins, hydrological skill scores (e.g. Kling-Gupta Efficiency or KGE) remain poor at event scales largely due to significant difficulties involved in estimating highly localized orographic precipitation in complex terrain (e.g. Mtibaa and Asano, 2022; Ghimire et al., 2022;





Arulraj and Barros, 2021; Barros and Arulraj, 2020; Zhang et al., 2012; Huffman et al., 2007;
Andrieu et al. 1997).

55          With global warming and projected heavier precipitation and higher probability of floods

in climate change hotspots and significantly modification of the hydrologic cycles in complex
terrain (Nijssen et al., 2001), substantial efforts have been made understand and quantify
precipitation uncertainties in the mountains and nearby lowlands (Pepin et al., 2022). Increased
warming causes reduction in snowpack during winter/spring impacting seasonal streamflow
patterns (Moraga et al., 2021; Saunders et al., 2008; Arnell 2003).  Increasing probabilities of
severe summer thunderstorms (e.g., Brooks 2013) are already one of the largest contributors to
global losses (> $10 billion USD a year, Allen 2018). The urgent need to improve precipitation
estimation and forecasting, particularly for warm-season flood-producing precipitation events, is
unprecedented.

Current approaches involved in precipitation measurement and Quantitative Precipitation

Estimation (QPE) broadly include in-situ point-scale observations using rain gauges and
disdrometers, and remote spatial observations using ground-based radar and space-based sensors.
In complex terrain, there is often a scarcity of  in-situ measurements due to difficult access. Radar-
based QPE products are plagued by uncertainties from various sources (e.g. ground clutter effects,
retrieval uncertainties, radar viewing geometry, (Villarini and Krajewski, 2010; Arulraj and
Barros, 2021; Barros and Arulraj, 2020; Zhang et al., 2012; Kreklow et al., 2020; Huffman et al.,
2007; Andrieu et al. 1997; Creutin et al., 1997; Durden et al., 1998)). Numerical weather prediction
(NWP) is an alternative to measurement.  However, QPE produced by NWP models are
characterized by large uncertainties when evaluated against rain gauges (e.g. Zhang and
Anagnostou, 2019) leading to large runoff deviations from streamflow measurements when used





as inputs to hydrological models (e.g., Tao et al., 2016; Weiland et al., 2015; Diomede et al., 2008;
Kobold and Suselj 2005). Due to these uncertainties and errors involved, significant efforts have
been devoted to improving QPE via various approaches in the past few decades such as radar-
raingauge data fusion (e.g. McKee and Binns, 2016; Goudenhoofdt and Delobbe, 2009; Delrieu et
al., 2014; Nanding et al., 2015; Sideris et al. 2013; Schiemann et al. 2011; Berndt et al. 2014),
radar reflectivity and retrieval corrections (e.g. Vignal et al., 2000; Shao et al., 2021; Dinku et al.,
2002) and data assimilation of various radar products (e.g. Rafieeinasab et al., 2015; Wehbe et al.,
2020). Rain gauge and disdrometer measurements often provide the ground truth for these QPE
optimization approaches (e.g. Harrison et al., 2000; Shao et al., 2021; Fulton et al., 1998). The
'ground truth', however, has its own error (e.g., wind effects around the gauge orifice,
Kochendorfer et al., 2017), and fails to capture highly localized orographic enhancement (e.g. Prat
and Barros, 2010; Gentilucci et al., 2021; Buytaert et al., 2006). Gauge-radar fusion often relies
on geostatistical assumptions that are primarily distance-based (e.g. Areerachakul et al., 2022;
Cassiraga et al., 2021; Wang et al., 2020; Maggioni and Massari, 2018), lacking the full picture of
basin topography which has a regulating role in orographic QPE.   Although there is no definite
consensus on guidance for the placement of rain gauges in mountainous basins (e.g. Suri and Azad,
2024), Barros et al. (2004), Prat and Barros (2010), Barros (2013), Barros et al. (2014), and Duan
et al. (2015) provide consistence guidance regarding the importance to install precipitation
networks on the topographic envelope and at regular intervals along ridges and adjacent valleys
using examples from the Central Himalayas, the Central Andes and the Southern Appalachians.

To address this long standing QPE challenge in complex terrain mainly, a general QPE

error quantification framework was developed leveraging widely available United States
Geological Survey (USGS) streamflow observations at the outlet of headwater basins in complex





terrain, consisting of 2 distinct pathways: 1) rain gauge bias correction, and 2) grid-level QPE
correction constrained to watershed-scale water budget closure. The first  pathway includes rain
gauge bias corrections at gauge locations both at diurnal and climatological scale, and
geostatistical distribution of raingauge biases across a basin. The second  pathway includes an
innovative inverse QPE correction method by backward propagating runoff uncertainty using a
hydrological model via streamlines to precipitation at storm-event scale (i.e. Inverse Rainfall
Correction or IRC, Liao and Barros, 2022 or LB22). It is also worth noting that runoff uncertainty
in hydrological modeling stems from various sources, generally including forcing uncertainties,
land surface condition uncertainties and model specific uncertainties (e.g. Clark, et al., 2008;
Beven and Binley, 1992).
LB22 found that initial soil moisture uncertainty can prevent the IRC framework from
achieving water budget closure because large initial condition errors cause significant travel time
uncertainties. Soil moisture is considered a particularly important factor among soil properties due
to its significant role in regulating runoff generation, hence dramatically altering flood timing and
magnitudes (e.g. Marchi et al., 2010; Penna et al., 2011; Yin et al., 2022; Vivoni et al., 2007), and
soil moisture can change very fast at sub-daily time scales changing saturation to nearly wilting
point levels depending on soil texture and land-use (Grillakis et al., 2016).  Initial soil moisture
conditions can therefore determine whether a rainstorm produces a major flash flood or not (Zehe
and Blöschl, 2004, Komma et al., 2007). However, due to the lack of in situ soil moisture sensors
and reliable high resolution soil moisture products, only a limited number of previous studies in
the peer-review literature focused on soil moisture uncertainty and the impact on flood simulation
(e.g. Laiolo et al., 2016; Zappa et al., 2011; Tao et all. 2016; Silvestro et al., 2019; Silvestro and
Rebora, 2014; Uber et al., 2018). Liao and Barros (2024b) developed an Initial Condition





Correction (ICC), based upon travel time theory, which is consistent with the general IRC
framework, demonstrating large improvements in initial soil moisture estimation. Note that when
implementing the IRC and ICC, we are using a fully distributed physics-based uncalibrated
hydrological model (i.e. Duke Coupled Hydrological Model, DCHM) that has been used for over
25 years with great success in the Southern Appalachians (e.g., Tao and Barros, 2013, 2014, 2016),
and consequently uncertainty from the model and model parameters is assumed to be negligible.
Hydrological model parameters have impact on rainfall-runoff response, but they are generally
only of secondary importance compared to the precipitation proper and initial soil moisture
conditions particularly in small basins (e.g. Mockler et al., 2016; Dobler et al., 2012; Zappa et al.,
2011). Liao and Barros (2024b) also demonstrate that QPE uncertainty dominates runoff
uncertainty for extreme precipitation events compared to initial condition uncertainty, and initial
conditions only begin to play an important role for less extreme events particularly early in the
event prior to the rapid rise of the hydrograph, which is consistent with previous studies where
initial soil moisture uncertainty usually has a decreasing impact on runoff uncertainty as
precipitation continues (e.g., Figure 8 in Iwasaki et al., 2020).
In this work, IRC and ICC are coupled into one framework (referred to as the coupled IRC-
ICC framework) to construct a high resolution QPE dataset aiming to close the water budget at the
scale of storm-flood events along the latitudinal range of the Appalachian Mountains across
diverse hydroclimatic and geomorphic regions. The coupled IRC-ICC framework is applied to 26
headwater basins and 215 flood-producing events from 2008 to 2024 using Next Generation
Weather Radar (NEXRAD) StageIV dataset as original inputs, at a spatial and temporal resolution
of 250 m and 5 minutes respectively, and the improved post IRC-ICC QPE data (i.e. StageIV-IRC)
are made available in this study.




The manuscript is organized as follows. The data sources, and QPE error quantification
framework which consists of raingauge bias correction and the coupled IRC-ICC framework, are
detailed in Section 2. Section 3 presents this new dataset (StageIV-IRC) along with data
assessment from various aspects. Section 4 discuss the potential application of this new dataset
and future work. Section 5 provides the access to the dataset and summary of the work.

## 2.  Data and Methodology


### 2.1 Radar QPE StageIV


The NCEP/EMC (Environmental Modeling Center) StageIV is a QPE product from the
National Weather Service (NWS) derived from the regional hourly and 6-hourly multisensor
(radar + NWS raingauges) precipitation analyses (MPEs), which is further improved with new
analyses from River Forecast Centers (RFCs) over the conterminous United States (CONUS) (Lin
and Mitchell, 2005). In mountainous regions, StageIV datasets suffer from the blockage of
complex terrain, resulting in ground clutter, overshooting and retrieval uncertainties,
demonstrating significant biases and errors in rainfall detection. In support of the NASA's
Precipitation Measurement Missions (PMM) program ground validation (GV) activities (Prat and
Barros, 2010a), a dense network of raingauges was installed in the Southern Appalachians in 2007
and has been well-maintained since 2007 (Barros at al., 2014). In this study, the raingauge
observations from the Southern Appalachians are used to correct StageIV.

### 2.2 GV Raingauge Observations


A high resolution raingauge network consisting of 34 tipping bucket raingauges has been
maintained in the Pigeon River basin, over the ten-year reference period 2007-2018, during and



immediately after IPHEx (Integrated Precipitation and Hydrology Experiment or IPHEx, Barros
et al. 2014). A map of the raingauge network is shown in Figure 1 with each raingauge labelled
with a number, and the detailed locations of these gauges are documented in Table 1. Besides
spatial representativeness errors related to the setup of the raingauge network as stated earlier,
common errors include funnel wetting, pipe clogging and turbulent winds near gauges (e.g. Wang
and Wolff, 2010). The raingauge network is regularly visited and maintained for at least three
times a year including on-site cleaning and calibration. In this work, we use these rainfall
measurements to adjust hourly StageIV QPE. In-situ rain-gauge data are publicly and available
and can be found at http://dx.doi.org/10.5067/GPMGV/IPHEX/GAUGES/DATA301.(Barros et
al., 2017) In addition to raingauges, a network of Parsivel disdrometers was installed during the
IPHEx EOP (Extended Observing Period, 2013-2014), with each disdrometer location denoted by
the letter P in Figure 1. Due to the limited duration of the disdrometer measurements, the
disdrometer data were used only for the purpose of independent evaluation. Note that raingauges
are placed mostly on ridges while disdrometers are generally located on the hillslopes and
lowlands.

<Figure 1 here please>

**2.3 Methodology**
The methodology of this work includes 3 major components: 1) raingauge bias correction,
2) grid-scale QPE correction by closing the water budget using streamgauge measurements, and
methods involved in 3) basin and event selection procedures, and model setup.

### 189   2.3.1 Raingauge Bias Correction

A schematic drawing of the raingauge bias correction framework to derive gauge-improved
QPE (named StageIV$_{DBKC}$) is summarized in Figure 2.
<Figure 2 here please>

First, to make meaningful comparison between StageIV and raingauges in space, a fractal
downscaling algorithm is used to generate high spatial resolution StageIV$_D$ at 1km from the
original StageIV product (4 km resolution). After downscaling, bias correction at event scale and
ordinary kriging are applied consecutively to modify the StageIV$_D$ to StageIV$_{DB}$ at hourly time-
scale. Subsequently, StageIV$_{DB}$ data are evaluated against the raingauge climatology from 2008 to
2017 to reduce any remaining biases conditional on weather regime, and climatological biases are
geostatistically interpolated using the ordinary Kriging method. The resulting dataset is named
StageIV$_{DBKC}$ (abbreviated as STIV$_{DBKC}$).

### 202   2.3.2 Fractal downscaling

Aiming to derive high resolution QPE datasets in complex terrain, the assumption of self-similarity
is imposed. The Hurst coefficient $H$, fractal dimension $D$, and the spectral exponent β are described as the
following:
$$D = \frac{7-\beta}{2} \tag{1}$$

$$H = \frac{\beta-1}{2} \tag{2}$$



The parameter β describes rainfall statistics at different scales, and it is estimated as the slope of
the power spectral density curve in 2D Fourier domain of the rainfall field (log-log plot). The power spectral
density Z(u,v) in the 2D Fourier domain is :

$$Z(u, v) = \left(\frac{L}{N}\right)^2 \sum_{x=0}^{N-1} \sum_{y=0}^{N-1} z(x, y) \, exp\left[-\frac{2\pi i}{N}(ux + vy)\right] \tag{3}$$

where u and v represent the transform of x and y in Fourier domain, N is the total number of grid
points in each direction, and z(x,y) is the rainfall field. Additionally, the spectral density at wavenumber k
= 1 is defined as the roughness factor, that is the variance of the field. The Hurst coefficient describes the
auto correlation strength (range from 0 to 1) with higher values of H implying higher auto-correlation, that
is persistence. The mean power spectral density in 2-D Fourier domain is given:

$$S_j = \frac{1}{L^2 N_j} \sum_{1}^{N_j} |Z(u, v)|^2 \tag{4}$$

where $N_j$ is the number of coefficients that satisfy the condition $j < \sqrt{u^2 + v^2} < j + 1$. The mean
power spectral density has a power-law relationship with wave number k, and k is defined as below:

$$k = \frac{2\pi}{\sqrt{u^2 + v^2}} \tag{5}$$

$$S \sim k^{-\beta - 1} \tag{6}$$

where $\beta$ is the spectral exponent calculated as the slope of power density spectrum. Assuming the
rainfall fields are self-similar, then the information at fine resolutions can be derived from the information
at coarser resolution. This is accomplished using a Brownian surface ($Z_b$, H=0.5) at the desired fine
resolution as spatial support for the interpolation, which is modified in the Fourier domain ($Z_D$) to replicate
the distribution of energy slope of the spectral slope and roughness factor according to Bindlish and Barros

(1996):

$$Z_D(u, v) = \frac{Z_b(u,v)}{k_r^{(\beta - \beta_b)/2}} \, exp\left[\frac{1}{2}\left(S_{r,1} - \frac{\beta + 1}{\beta_b + 1} S_{r,2}\right)\right] \tag{7}$$





where $\beta$, $\beta_b$, $Z_D(u,v)$ and $Z_b(u,v)$ are respectively the original rainfall fields spectral exponent,
Brownian surface spectral exponent, interpolation surface in Fourier domain and Brownian surface,
respectively; $k_r$ is the wavenumber and $S_{r,1}$ and $S_{r,2}$ are the roughness factor of the original rainfall fields
and Brownian surface. Due to the non-uniqueness of Brownian surfaces, multiple replicates of interpolation
surfaces $Z_D$ can be obtained. In this study, an ensemble of ND interpolation surfaces is derived, thus ND
rainfall fields at finer resolution preserving the same rainfall statistics at coarse resolution are generated.
Following Nogueira and Barros (2015), here ND=50 and the correction steps described in Figure 2 are
applied to the ensemble mean of the downscaled rainfall fields.

### 2.3.3 Bias Correction

The *first* phase of bias correction is carried out at event scale: a linear regression is
established between raingauge measurements and collocated downscaled radar pixel estimates
using the following formula:
$$R_g^t(i_g, j_g) = \kappa R_r^t(i_g, j_g) + \varepsilon \tag{8}$$

where $R_r$ represent radar measurements, $R_g$ represent raingauge observations, $\kappa$ and $\varepsilon$ are
respectively the slope (conditional bias correction) and the intercept (systematic bias correction).
For each hour, collocated pairs of StageIV$_D$ estimates and raingauge observations within a radius
of 5 km centered on the StageIV$_D$ pixel were identified as long as more than two raingauges
measure non-zero rainfall. Regional least-square linear regression was applied subsequently to all
StageIV$_D$ pixels within $\pm 1$-$\sigma$ deviation of the regional regression line at hourly time-scale by
assuming homogeneity of variances or homoscedasticity.
The *second* phase of bias correction is done at climatological scale: aiming to reduce
systematic radar errors caused by retrieval uncertainties and viewing geometry in complex terrain,
demonstrating strong diurnal (time of day) and seasonal (weather regime) error dependencies when



comparing against 10-year raingauge observations due to miss detection of shallow rainfall related
to radar overshooting in the Southern Appalachian (e.g. Wilson and Barros, 2014; Duan and
Barros, 2017; Arulraj and Barros, 2017). For this purpose, the following corrections were added
for rainfall below and above a threshold X, where X=2mm/hr in the Pigeon River Basin. When
raingauge measurements are less than 2mm/hr and Stage $IV_D$ estimates are 0, the StageIV$_D$ value
was replaced by the raingauge observations, here termed Light Rainfall Correction (LRC).
Furthermore, for each hour, nil StageIV$_D$ estimates where raingauge measurements are greater than
X=2mm/hr were identified and replaced by the mean of the corresponding collocated raingauge
measurements, hereafter Mean Rainfall Correction (MRC). Finally, for localized precipitation (i.e.
only two or fewer rainguages detect nonzero rainfall) normally associated with isolated convective
activity, the anomalies calculated as the differences between the StageIV$_D$ and the local raingauge
measurements were linearly distributed among the surrounding 25 pixels (5 pixel window centered
at the StageIV$_D$ pixel)– Convective Rainfall Correction (CRC).When more than 2 raingauges
measured rainfall, then the anomalies for each pixel were spatially distributed using ordinary
Kriging as described below – Global Rainfall Correction (GRC).
**2.3.4 Ordinary Kriging**
The Ordinary Kriging (OK) is a linear weighted geostatistical estimator that interpolates values of
a variable at a specific location using weights derived from spatial covariances aiming to minimize
prediction variance. In our case, the rainfall differences among raingauge measurements and StageIV$_{DB}$ at
all raingauge locations were calculated and denoted as $G(x_i)$ at gauge location i. To produce estimates at
any location within the study domain, a continuous model describing spatial covariance structure of the
data is necessary. A commonly used semi-variogram model is the spherical model, exhibiting linear
behavior at the origin. A review and comparison of different types of semivariogram models can be found
(e.g. Li and Heap 2008; Oliver and Webster, 2014; Zimmerman and Zimmerman, 2012). Bohling (2005)



analyzed the differences of several commonly used semivariogram models and pointed out that, given the
same variogram parameters (nugget, sill and range), spherical models reach to the maximum for
comparatively shorter spatial lags, and thus are suitable to capture strong spatial dependencies over short
distances as in the case of orographic precipitation (see also McBratney and Webster, 1986, for detailed
description of spherical model):
$$\gamma(h) = C_0 + (C - C_0)\left(\frac{3h}{2d} - \frac{1}{2}\left(\frac{h}{d}\right)^3\right) \text{ if } 0 \leq h \leq d \tag{9.1}$$

$$= C \qquad\qquad\qquad \text{if } h > d \tag{9.2}$$

$$\gamma_{0i} = \frac{1}{N_A}\sum_{k=1}^{N_A}\gamma_{ki} \tag{9.3}$$

$$\gamma_{00} = \frac{1}{N_A}\sum_{k=1}^{N_A}\sum_{l=1}^{N_A}\gamma_{kl} \tag{9.4}$$

where d is the range, h is the lag, $N_A$ is the number of available gauge locations, C and $C_0$ are the sill and
nugget values. Neglecting local variability and measurement error at the spatial scale of the downscaled
radar and raingauge (point) measurements, the nugget is constant and equal to zero (Diggle & Ribeiro,
2007). The rainfall difference at a target point $x_0$ $Z_{ok}^*(x_0)$ is calculated using a weighted linear combination
of all available differences with constraints of unbiased estimator
$$Z_{ok}^*(x_0) = \sum_{i=1}^{n}\lambda_i^{ok}G(x_i) \tag{10.1}$$

$$\sum_{i=1}^{n}\lambda_i^{ok} = 1 \tag{10.2}$$

Optimal weights can be obtained by solving following equation by employing Lagrange multiplier $\mu$:
$$\begin{pmatrix} \gamma_{11} & \cdots & \gamma_{n1} & 1 \\ \vdots & \ddots & \vdots & \vdots \\ \gamma_{1n} & \cdots & \gamma_{nn} & 1 \\ 1 & \cdots & 1 & 0 \end{pmatrix}\begin{pmatrix} \lambda_1^{OK} \\ \vdots \\ \lambda_n^{OK} \\ \mu \end{pmatrix} = \begin{pmatrix} \gamma_{01} \\ \vdots \\ \gamma_{0n} \\ 1 \end{pmatrix} \tag{11}$$

In this study, OK distributes spatially the differences between available raingauge measurements and radar
data, resulting in the generation of STIV$_{DBKC}$ dataset.



**2.3.5 Ordinary Kriging**

Standard performance metrics (McBride and Ebert 2000; Wang, 2014) including false alarm rate (FR), probability of detection (PD), threat score (TS), and Heidke skill score (HSS), as well as bias, and the root-mean-square error (RMSE) are used to evaluate the corrected downscaled hourly rainfall. An instance when both radar QPE and raingauge observation exceed a specified rain rate threshold is a hit (H); when observation matches the criterion and radar QPE does not, it is classified as a miss (M), if the opposite happens, then it is a false alarm (FA). The performance metrics are determined by combination of Hs, Ms and FAs:

$$Bias = \frac{1}{N}\sum_{n=1}^{N}(O_n - R_n) \tag{12}$$

$$RMSE = \sqrt{\frac{1}{N}\sum_{n=1}^{N}(O_n - R_n)^2} \tag{13}$$

$$FR = \frac{FA}{H+FA}, 0 \leq FR \leq 1 \tag{14}$$

$$PD = \frac{H}{H+M}, 0 \leq PD \leq 1 \tag{15}$$

$$TS = \frac{H}{H+FA+M}, 0 \leq TS \leq 1 \tag{16}$$

$$HSS = 2 * \frac{Z*H - FA*M}{((H+FA)*(Z+FA)) + ((M+H)*(M+Z))}, -1 \leq HSS \leq 1 \tag{17}$$

where O is raingauge observation and R is radar QPE, and N is the number of points. Z is the overall number of zeros (when neither radar data nor raingauge measurements match the threshold criterion). A TS of 0.5 implies that the criterion is satisfied at least 50% of the time, and a higher value is indicative of superior performance. A TS=0.33 is indicative of performance similar to persistence, meaning predicted values in the next hour are the same values in the previous hour. HSS describes the fractional improvement of the corrected STIV$_{DBKC}$ over original StageIV. An HSS of 0 means that the performance is not better than random chance.

**2.3.6 Hydrologic Correction**

At flash flood timescale in headwater basins, streamflow uncertainty and precipitation uncertainty are strongly connected in a nonlinear way through rainfall runoff processes. Liao and Barros (2022) developed a Lagrangian-based framework named Inverse Rainfall Correction (IRC) allowing backpropagating streamflow uncertainty to precipitation inputs in space and time through an uncalibrated distributed hydrological model (i.e. DCHM), achieving water budget closure at event scale in small headwater basins. As stated earlier, the uncertainties associated with parameters and the hydrological model DCHM are neglected since the model configurations have been used and improved over the past two decades for this region accounting for various soil, vegetation, and river processes (e.g. Tao and Barros, 2013, 2014, and 2018; Lowman and Barros, 2016), and the IRC framework has been tested in multiple headwater basins extensively in this region with consistent success.

It is worth noting that IRC is a general framework to improve QPE at watershed scale that can be incorporated into any distributed hydrological models. Liao and Barros (2024a, 2024b) investigated the impact of model structure uncertainty and initial condition uncertainty on IRC and then the downstream product: the resulting IRC-improved QPE. The results suggest with improved watershed physics at finer resolution (e.g. river bank storage, Liao and Barros, 2024a), river routing algorithms (e.g. XY routing, Liao and Barros, 2024a) and improved antecedent soil moisture distributions (Liao and Barros, 2024b), post-IRC QPE demonstrate much more realistic precipitation features at high resolution that are aligned with basin topography with ridges associated with higher precipitation than valleys in general, showing a significant improvement from the original StageIV dataset which is characterized by unnatural boxy precipitation patterns in complex terrain due to resolution issue.





As briefly mentioned before, LB22  reviewed various sources of uncertainty that can
prevent post-IRC QPE from achieving water budget closure, among which initial condition
uncertainty in soil moisture is a noteworthy source. Improved initial condition estimation results
in significantly improved post-IRC precipitation features in complex terrain by better capturing
transient travel time distributions (Liao and Barros, 2024b). They found that the uncertainty tied
to initial conditions is relatively more important for less extreme events. Nevertheless, the initial
condition correction (i.e. ICC) method is coupled with the IRC framework and the complete
framework is named the coupled IRC-ICC framework since Liao and Barros (2024b) to reflect the
importance of Initial Condition Correction (or ICC). The specifics of IRC, ICC and the coupled
IRC-ICC framework are schematically drawn in Figure 3.

<Figure 3 here please>

Based on the characteristic timing definitions in panels d) and c), different temporal
windows are identified corresponding to different flow regimes. In principle, many more windows
can be identified if a rather complex hydrograph with more peaks and inflection points is presented.
ICC is only applied to window 2 and 5 (i.e. rising point mismatch segment, and slow recession,
respectively), assuming precipitation uncertainty is dominating streamflow differences for window
3 and 4 (i.e. the neighborhood of flow peak). $W_{nm}$ represents precipitation after window m
procedure at iteration n. DCHM stands for Duke Coupled Hydrology Model, which is an
uncalibrated physics-based distributed model, and the spatial and temporal resolution are 250m
and 5 min.





**2.3.7 Lagrangian Tracking**
A flood event is simulated by the DCHM to simulate streamflow at the basin outlet with
grid-based time-varying velocity fields for different soil layers. When the precipitation starts (i.e.
basin-averaged precipitation > 0.1mm/hr), new particles (passive tracers)  are launched at non-
zero precipitation grids at *every* model time step (i.e. 5 minutes) in all soil layers following the
velocity fields calculated by the DCHM, and the tracking resolution is 10 seconds, amounting to a
release of approximately 600,000 particles for basin with an area of 120km$^2$ over a 24-hour period.
During the tracking phase, each particle is saved along with information regarding its source
location (grid-point where it originates), time of release ti, and travel time tT (tT is defined as the
difference between current time t and the time of release ti, i.e., tT = t – ti ). Multiple particles from
different source locations can have the same travel time, which is the basis for identifying the
number of trajectories contributing to the hydrograph at the outlet as a function of time.
**2.3.8 QPE Correction**
At time t, the water difference *wd(t)* between the observed and simulated streamflow over
the time *Δt* between two consecutive discharge observations represents the fraction of runoff that
eventually leave the basin as streamflow. Errors in precipitation forcing propagate to the runoff,
under the assumption of negligible model and parameter uncertainties, *wd(t)* can be entirely
attributed to precipitation error, which is the focus of this work.
$$wd(t) = [Q_{obs}(t) - Q_{simu}(t)] \times \Delta t \qquad (18)$$
The subscripts *obs* and *simu* refer to observed and simulated discharge, respectively.
The strategy for the inverse rainfall correction (IRC) using hydrograph analysis is to follow the
trajectories available from the Lagrangian tracking backward from the basin outlet to the source




locations at time $t_i$ and apply a correction at the source locations proportional to original QPE
magnitude to reduce $wd$ at time $t$. The embedded assumption is that larger QPE values have larger
uncertainties. Note that QPE corrections happened earlier in time will have an impact on runoff
simulation at future times, and this is the reason why IRC framework is a recursive framework.
The detailed rainfall correction steps can be found in (Liao and Barros, 2022).

**2.3.9 Methods for Reducing Uncertainties from Other Sources**

As briefly mentioned before, uncertainties from other sources (e.g. river routing, model

physics, antecedent soil moisture, etc) have certain impacts on travel time distributions and
simulated streamflow. Previous studies demonstrate that, for flood producing events in small
headwater basins, streamflow response is largely controlled by precipitation inputs (e.g. Iwasaki
et al., 2020). In this section, we briefly describe the methods used to minimize the impacts from
other sources to facilitate the IRC framework to achieve water budge closure.

First and foremost, the DCHM is an uncalibrated model with parameters strongly tied to

this region of study, demonstrating great success over the past 25 years. DCHM has been used
extensively without significant biases, therefore parameter uncertainty and model structure
uncertainty are ignored. The impact of routing algorithm on peak flood timing is investigated in
Liao and Barros (2024a) and they pointed out that variable parameter Muskingum-Cunge routing
leads to artificially fast rising limb of flash flood hydrographs in headwater basins due to the
existence of mild slopes over the valleys. They developed a general routing framework which is
more suitable for stream routing particularly for estimating flood timing in headwater basins (see
Liao and Barros, 2024a for details). Their results also suggest meandering effects, riverbank
storage, and initial soil moisture distributions can impact the early rising period of the hydrographs.
Later, significant and consistent improvements are made when introducing an initial condition





correction (ICC) module to reduce initial condition uncertainty (Liao and Barros, 2024b). In fact,
numerous studies also point out that precipitation and initial condition are the 2 most important
factors in hydrologic forecasting and simulation. Therefore, this innovative ICC module is coupled
with the IRC framework since then. The red arrows in Figure 3e indicate where ICC are executed
in the general architecture of the IRC framework and the specifics of the ICC module are stated
below.

Particles launched during the IRC process that reached the outlet at time t are traced back

directly to the IC timing or time 0, and their locations at the IC timing are shown in the bottom
maps in Figure 3d as the control points of time t. The downstream area of the control points has
shorter transportation time to arrive at the outlet (e.g. water difference $\Delta S_1$), and upstream area of
the control points takes longer to get to the basin outlet (e.g. water difference $\Delta S_2$). Similarly, soil
moisture in the impacted area can greatly impact the size of $\Delta S_2$ and flow conditions after the
timing $t_2$. Assuming initial condition are only impactful during early period and late recession of
hydrograph, which is supported by the fact that these events are flood-producing events with large
QPE uncertainties dominating the vicinity of peak flow, ICC is used for hydrological windows
near the peak flow. Following the same notation in the IRC framework (Eq. 18), using backward-
in-time numerical notation, $wd(t)$ represents the flow volume difference between simulated and
observed flow between time $t$ and $t - \Delta t$. A 'band' of region can therefore be identified, that is
the region downstream to control points of time $t$ and upstream to control points of time $t - \Delta t$.
This 'band' is then referred to as the impacted area for time $t$, and the initial soil moisture in the
impacted area significantly influence basin discharge between time $t - \Delta t$ and time $t$. Finally,
$wd(t)$ is then applied to initial soil moisture within the impacted area (i.e. the 'band') and the
details can be found in Liao and Barros (2024b).





**2.3.10 Hydrological Skill Metrics**
The hydrological skill metrics used in this study include the Kling-Gupta Efficiency (KGE)
of the streamflow calculated at time-interval τ (here 15 minutes) determined by the frequency of
observations (i.e. USGS gauge records) over the event duration (here 24 hours):
$$KGE_\tau = 1 - \sqrt{(r-1)^2 + (\frac{\sigma_{sim}}{\sigma_{obs}} - 1)^2 + (\frac{\mu_{sim}}{\mu_{obs}} - 1)^2}$$
(19)

where r is the correlation, $\sigma_{obs}$ is the standard deviation in discharge observations, $\sigma_{sim}$ the
standard deviation in discharge simulations, $\mu_{sim}$ and $\mu_{obs}$ are the mean streamflow of the
simulations and the observations respectively. The subscript τ denotes the time-scale dependence of
the KGE. KGE ranges from negative infinity to 1.
The Nash Sutcliffe Efficiency (NSE) of the streamflow is also calculated at time-interval τ
(here 15 min):
$$NSE_\tau = 1 - \frac{\sum_{t=1}^{T}(Q_o^t - Q_s^t)^2}{\sum_{t=1}^{T}(Q_o^t - \overline{Q_o})^2}$$
(20)


where $Q_o^t$ and $Q_s^t$ are the streamflow observation and simulation at time t, and t is ranging from the
first time step to the last time step T. $\overline{Q_o}$ is the mean of observed streamflow. The subscript τ denotes
the time-scale dependence of the NSE. NSE ranges from negative infinity to 1.
The relative volume error (EV) is the difference between the time integral of the simulated
and observed hydrographs over the event discharge volume:



$$EV = \frac{\text{Simulated hydrograph volume} - \text{Observed hydrograph volume}}{\text{Observed hydrograph volume}} \qquad (21)$$


An overestimation case has EV>0 and an underestimation event has EV<0.
EPT is the error in the timing of peak discharge on the rising limb of the hydrograph. When
calculating EPT, if multiple peaks are present, only the highest peak timing is considered. To better
capture the difference between rising limbs of observations and simulations, EPT is calculated
using both the rising point and the highest peak point; thus, EPT compares the difference between
the mid points of the two rising limbs.
EPV or error in peak volume is a relative error between simulated and observed flood peak,
and the equation is below:
$$EPV = \frac{\text{Simulated peak flow} - \text{Observed peak flow}}{\text{Observed peak flow}} \qquad (22)$$


**2.3.11 Study Domain and Model Setup**
28 headwater basins are selected in the Appalachians as illustrated in Figure 1 with basin
drainage area ranging from 50 km$^2$ to 500 km$^2$. These headwater basins cover a wide range of
geographic regions (e.g. Basin01 and Basin30 are over 2,000 km apart) with diverse weather and
climate regimes, associated with large differences in geomorphology and hydrogeology.

<Figure 4 here please>






Soil-related parameters are downloaded from a global high resolution (1 km) soil data
repository (Zhang et al., 2018). For each basin, the vertical hydraulic conductivity remains the
same for the entire soil column. The lateral hydraulic conductivity in the unsaturated zone was
assumed to be two-three orders of magnitude larger than the vertical conductivity in the shallow
soil layers, with higher values where the stone fraction in the soils is higher (Carlson, 2010, Freeze
and Cherry, 1979). The final scaling factors were obtained through simple sensitivity analysis to
match the curvature and slope of the observed subsurface runoff recession curves (Yildiz and
Barros, 2007; Chen and Kumar, 2001; Linsley et al., 1982), and the final scaling factors are: 1500,
150, 15 and 1.5 for layer 1 (0-10 cm), layer 2 (10-75 cm), layer 3 (75-200 cm) and layer 4 (2-20
m), respectively. No parameter calibration is done in this work as the primary focus of this work
is to develop a QPE dataset that can consistently close the water budget while controlling
uncertainties from other sources, largely advancing the understanding of QPE uncertainties across
climate, weather, and geomorphological regimes.
Flood-producing events are selected for the headwater basins identified in the
Appalachians for recent years (i.e. the study period is from January 2021 to April 2024). The
selection criteria are threshold-based, specifically the peak flow must be greater than 95% of the
flow records in the study period. The choice of 95% is a compromise because 99% would yield
too few events while 90% would be too close to the annual flood. Additionally, rainfall runoff
response time must be shorter or equal to 6 hours to be qualified as a flash flood event. Only warm
season liquid precipitation events 2021-2024 are finally selected during this systematic event
selection process. Here, the warm season is specifically defined as from April 1st to September



30th. Note, data quality control is enforced and events with missing streamflow records are
discarded.
For the Cataloochee Creek Basin (Basin05), located in the SAM known to have
experienced multiple flash floods in the past (Tao and Barros, 2013), Liao and Barros (2024a)
created a Historical Flood Record database (HFR) for this basin, which includes numerous flood-
producing events from 2008 to 2017. The event selection criteria when developing HFR are also
using the same 95% flow threshold method. The difference is that the HFR also includes multiple
winter-time liquid precipitation events that result in flash floods. In total, there are 54 events for
Basin 05 in the HFR and these events are also used to expand the study sample size in this work.
The hydrological spin-up process starts with iterative DCHM runs from April 30[th] to
September 30[th], 2021, including a total of 5 iterations (i.e. reaching a stable/equilibrium model
state). Then DCHM runs continuously from October 1[st] 2021, to April 1[st], 2024 to derive initial
conditions for events happened after September 30[th] 2021. During this spin-up process, no
parameter calibration is involved. The initial conditions used for the events in this study are from
the 5[th] iteration from April 30[th] to September 30[th], 2021, and from the subsequent continuous run
from October 1[st], 2021 to April 1[st], 2024.
**2.4 Caveat**
In the entire study domain, rain gauges are only installed in the Southern Appalachians
specifically in the vicinity of the Cataloochee Creek Basin (Basin 05) in the core area of the IPHEx
rain gauge mesonet. The remainder of the studied basins are not monitored by raingauges, and
therefore no raingauge bias correction is done for those basins and the downscaled original dataset
StageIV (i.e. STIV$_D$) is used as inputs for the IRC method and hydrological simulations in this
study.





As an important component of the IRC framework, the Lagrangian tracking algorithm is
only implemented when transitioning from one hydrological window to the next window, instead
of being implemented every model time step (i.e. 5 minutes), and this is because of computational
constraints. Additionally, we do not differentiate peak flow points and recession inflection points
between simulations and observations when classifying hydrological flow regimes/windows, and
consistently use observations as the reference basis for hydrological window identification because
1) precise locations of particles become much more uncertain later in the hydrograph due to
numerical rounding errors and grid-based abruptly-changing velocity fields used in the Lagrangian
tracking algorithm, and 2) the computational costs associated with excessive running of the
tracking algorithm. Very short travel times (i.e. <15 minutes) are ignored because of temporal
resolution restrictions from streamflow observations. A systematic use of 24 hours for event total
duration is imposed in this work to reduce excessive tracking workload, which might be
problematic for events with very long and heavy tails though not common for flash flood events
in headwater basins.
Nevertheless, the coupled IRC-ICC recursive framework allows us to quantify QPE
uncertainties more realistically by improving initial soil moisture estimation, and this framework
proves to be numerically efficient in achieving good and stable hydrological state after only a few
iterations. In this work, the stable state of IRC-ICC is reached when the Kling-Gupta Efficiency
(KGE) oscillations are within 0.05 from iteration to iteration, calculated at 15-minutes intervals.





## 3. Results and Discussion

### 3.1 Raingauge Bias Correction

Raingauge bias correction includes linear bias correction for radar-gauge pairs (see Eq. 8) and a series of biases corrections listed in Section 2.3.1: LRC, MRC, CRC and GRC. Analysis of the diurnal cycle on a seasonal basis reveals bias patterns linked to radar operations, and in particular terrain blockage, radar beam overshooting, and excessive attenuation that may vary from hour to hour but when taken over a long period of time indicate localized errors in space and time that reflect the site hydrometeorology. Light and shallow rainfall is a particular challenge in the region of study (e.g. Duan et al. 2015; Duan and Barros, 2017; Arulraj and Barros, 2017). The peak number of missed rainfall corresponds to about 10-15% of the total number of hours for each season in the late afternoon. The missed events correspond to both light and moderate rainfall, and occasionally to isolated heavy rainfall likely associatd with isolated thunderstorms.

The climatologically corrected $STIV_{DBKC}$ fields have significantly accurate diurnal cycle comparing to only event-scale bias corrected $STIV_{DBK}$. This processes is illustrated in Figure 5 for one raingauge in eastern ridges (left panel) and another in the western ridges (right panel).

<Figure 5 here please>

Biases in original $StageIV_D$ are more significant over the western ridges (e.g. right panel) at all times of day reflecting the impact of cloud immersion and seeder-feeder enhacement of low level precipitation over the ridges (Duan and Barros, 2017), with mid-day bias being a problem across the region (e.g., Barros and Arulraj, 2019). Overall, analysis of the $StageIV_{DBKC}$ fields



demonstrates that the climatology corrections work well in terms of mean rainfall, as well as
reducing miss detection errors. Figure 6 shows the diurnal cycle of missed precipitation at two
selected gauge locations (top row) in the winter (Januray-February and March – JFM) in StageIV
that are preserved in StageIV$_D$ (black) and StageIV$_{DBK}$ (cyan). These missed precipitation events
correspond to instances of very light rainfal (bottom row) at the raingauge locations (< 1.5 mm/hr).
After applying the LRC and MRC climatology corrections, the missed detection problems (cyan)
in StageIV$_{DBK}$ are largely eliminated for the StageIV$_{DBKC}$ fields (green).

<Figure 6 here please>

When integrated over the ten-year period, the averaged seasonal HSS, TS, and RMSE

statistics of STIV$_{DBKC}$ demonstrate significantly better performance comparing to STIV$_D$ for all
hours of the day (Figure 7a). Moreover, note that there is no decreasing trend in TS with increasing
precipitation rate threshold (Figure 7b) which indicates that climatology correction is working for
the heavy rainfall amounts linked to localized thunderstorm activity. Figure 7c shows the diurnal
cycle and seasonality distribution of RMSE conditional on rain rate. The RMSE generally stays
below 0.1 mm/hr except in the early morning and in the late afternoon in the cold season.  In part
this error could be related to snowfall which is not properly accounted for as the raingauges are
not heated.

<Figure 7 here please>





**3.2 Hydrologic Correction**

The coupled IRC-ICC framework is originally developed and applied in Basin 05, the Cataloochee Creek Basin, and an example showing the results from iterations is demonstrated in Figure 8. The notation follows the definition in Figure 3. Note $STIV_{DBKC}$ data derived in Section 3.1 are further downscaled to 250m and used for hydrological simulations in this section. For all other basins (except Basin05), raingauges are not available and $STIV_D$ data are used instead.

<Figure 8 here please>

It is demonstrated that for this extreme flood-producing event, IRC-ICC produces stable results after about 3 to 4 iterations without significant oscillations. In general, for less significant events, IRC-ICC often reaches an equilibrium state faster (merely 3 iterations), providing fast and convergent corrections. As explained in the Section 3, the equilibrium state is considered when oscillations in the KGE values are within 0.05, and then IRC-ICC is stopped immediately. This study suggests that for most events 3 iterations is a good rule of thumb.

**3.2.1 Systematic Application of IRC-ICC**

Systematic application of the coupled IRC-ICC framework is conducted in the 28 basins selected in the Appalachians for 225 events, and examples are displayed in Figure 9.

<Figure 9 here please>





The performance of IRC-ICC is in general slightly better in the Southern and Northern
Appalachian Mountains (SAM, NAM) than the Central Appalachian Mountains (CAM). In CAM,
particularly along the border of the state of West Virginia and the state of Virginia, residing
expanded karst terrain, and numerous caverns are identified (see the documented caverns:
http://www.wvgs.wvnet.edu/www/geology/docs/WV_Tax_Districts_Containing_Karst_Terrain.
pdf). The current version of DCHM hydrological model does not solve physics involved in Karst
terrain. Here, the advantage of not calibrating parameters becomes obvious because these Karst
terrain related physics can easily be ignored by parameter calibration when domain knowledge is
not sufficient. Being in Karst terrain, Basin 13 and 14 (not shown) demonstrate noticeably poor
simulations with severely underestimated baseflow contribution and artificial peaks due to the lack
of subterranean river representation. This is apparently beyond the resolved scale using the current
DCHM model with current spatiotemporal resolutions (250m, 5minutes). Here, resolved scale
refers to a reasonable scale range where physical processes are represented in the hydrological
model. At coarse scales, physical processes are substantially averaged, and information is
potentially lost during averaging. At fine scales, some physical processes are not yet known or not
parameterized in the model. The 10 events in Basin 13 and 14 are therefore discarded.
Event 2021-06-10 in Basin 19 (Figure 9) is an example when more hydrological windows
(see Figure 3) are required to capture the subtle changes in the hydrograph for a relatively more
complex hydrograph (e.g. multiple mild peaks around the major peak). These subtle changes could
be a shifting of dominant river branch in the basin due to the movement of rainfall. Again, this
requires much finer resolution both for the hydrological model and for the tracking algorithm to
represent this detailed level of physics for summer thunderstorms. With limited computational
power, this study systematically uses a 4-window IRC-ICC framework, including pre-rising-point



segment, rising limb, early recession, and late recession (separated by the recession inflection
point).

### 3.2.2 IRC and IRC-ICC Precipitation Corrections

Event total precipitation fields are calculated after IRC-only and IRC-ICC frameworks
reaching an equilibrium state, and these fields are compared with product STIV$_{DBKC}$ used as inputs
for these frameworks. Examples are shown in Figure 10 categorized by seasons in the Cataloochee
Creek Basin (Basin05). Again, warm season is defined as April 1$^{st}$ to September 30$^{th}$, and the rest
is defined as the cold season, with only liquid precipitation events are studied in this work.

<Figure 10 here please>

The original QPE (**a1** and **b1**) shows boxy patterns and abrupt spatial changes, which is a
common issue of radar observations when used at high spatial resolution. By contrast the IRC-
corrected precipitation maps (from both the IRC-only framework and the coupled IRC-ICC
framework) are aligned better with terrain gradients, showing strong spatial patterns with higher
precipitation along ridges and lower precipitation in adjacent valleys. IRC-ICC precipitation fields
have similar patterns to IRC-only precipitation. Note the dark blue colors corresponding to very
low precipitation near the basin outlet are an artifact of the IRC tied to very short travel times that
cannot be fully resolved even at the fine scales of 250m and 5minutes. However, with proper IC
uncertainty reduction, these artifacts are dramatically reduced as shown for the 2009-10-14, 2009-
04-20, and 2013-04-12 events because of overall basin-wide travel time improvements attributed





to improved IC. These three events are relatively mild events, indicating larger importance of IC
for events of lower magnitude because of the critical role of IC in runoff generation mechanisms
and travel times distributions.
**3.2.3 Precipitation and Hydrologic Statistics**
Event-total precipitation maps are calculated for each basin and event, and basin scale
precipitation statistics (e.g. mean and standard deviation) are derived for each event-total
precipitation map. These statistics are plotted in Figure 11, and subregions are separated by vertical
black lines. Basins 01 to 11 are located in the SAM, Basins 12 to 20 are located in the CAM, and
Basins 21 to 30 are located in the NAM. Given the dramatic impact of the Karst terrain on the
hydrological performance related to Basin 13 and 14, the results from these two basins are not
included in the statistics.
<Figure 11 here please>
It is clearly demonstrated that the change in the mean (i.e. basin-averaged event total QPE)
is relatively small (from 36.10mm to 38.07mm) compared to the change in the standard deviation
(from 6.63mm to 14.08mm) after the application of the coupled IRC-ICC. The small standard
deviation of the original QPE suggests that original QPE data are spatially tightly clustered with
low variability (see Figure 10a for boxy rainfall features), while larger standard deviation post-
IRC-ICC indicates spatial variability is enhanced, which is highlighted by the terrain-aligning
precipitation features in Figure 10c. The relatively small change in the mean indicates that the
original input precipitation (i.e. StageIV$_{DBKC}$ for Basin 05, and StageIV$_D$ for the remainder basins)
does not contain significant unconditional systematic biases across basins and events, which
would lead to consistent positive or negative flood volume errors. This argument is supported by
the fact that only small changes in the mean are introduced by the IRC-ICC framework. As an





exception, it is worth noting that the standard deviation of Basin 05 events does not change
significantly after the IRC-ICC compared to other basins and events because rain gauge corrections
are employed in Basin 05 but not anywhere else. It can never be overly emphasized that even after
rain gauge bias correction, essentially as a point-scale correction method, the resulting QPE is still
subjected to large water budget closure errors (see Figure 12 for more discussion)  on account of
the highly heterogeneous nature of QPE in complex terrain.

The hydrologic statistics described in Table 1 using all studied events are plotted in Figure

12.

<Figure 12 here please>

Figure 12 shows that the median KGE in each sub-region across the basins and events is

improved from 0.36, 0.39, 0.27 to 0.89, 0.74, 0.84 for SAM, CAM and NAM, respectively. It
should be pointed out that QPE changes for Basin 05 events (event number 55 to 108) are important
for improving water budget closure, albeit small in magnitude compared to other events in other
basins as shown in Figure 11 and 12, and yet critical to capture the complex precipitation
heterogeneity in complex terrain to close the water budget. Basin 05 is a good example illustrating
not only the contributions but also the limitations of rain gauge bias corrections in complex terrain
in general. The relatively mild improvement in the CAM is explained by lacking physics
representation of subterranean rivers in the Karst terrain in the DCHM model, causing large
baseflow errors during hydrograph recession and thus low KGE values. Nevertheless, for
applications in flash floods research, peak flood discharge, flood peak timing, and flood volume
are the most important factors (see the second, third and fourth horizontal subplots in Figure 12.
Flood volume error (the second panel) is controlled within ±10% for over 90% of the flood-
producing events in the Appalachians, with the median EV error being less than 5% for post IRC-



ICC products in SAM and NAM. Flood peak volume (the third panel) is controlled within 20%
for most of the events, which is significant because these events are extreme events. This is
demonstrated by Tropical Storm Fred on 2021-08-17 and event that caused floods in multiple SAM
basins, caused 5 deaths and an estimated economic loss of over 1 billion dollars: the KGE improves
to 0.9 and peak timing errors are less than 30 minutes using IRC-ICC. For most of the studied
events, timing errors (shown in the fourth panel) of the post IRC-ICC product are bounded by ±60
minutes, though some outliers are observed in the CAM and NAM potentially due to complex
surface conditions such as antecedent snow on the ground for April events.
Events with relatively large timing errors (±90 minutes) are investigated in detail. These
include the 2023-07-08 event in Basin 27 in New Hampshire (event number 185, which is 2.5
hours too early). This was a localized summer thunderstorm event, only taking half an hour to
reach its peak flow, posing a challenge in separating flow regimes using hydrological windows
defined in the IRC-ICC framework at the current model resolution. The event on 2022-05-27 in
Basin16 (a relatively big basin >400 km$^2$) is a relatively slow rising event (event number 118, 2
hours too early) in West Virginia with rain-on-snow conditions and potentially snowmelt effects
involved at high elevations. Finally, event 2021-09-22 in Basin19 (event number 133, 2 hours too
late) is a relatively more complex event with multiple rain cells moving over the basin close to
each other in time, therefore requiring many more windows to capture highly transient
hydrological regimes than the 4-window default structure (i.e. pre-rising limb, rising limb, early
recession, late recession) used in the IRC-ICC.
Overall, there are significant improvements in QPE corresponding to improvements in
flood volume, flood peak and flood timing as a result of IRC-ICC. Because the IRC-ICC is a
framework built upon runoff travel time, it cannot be used when precipitation is missing or there



are  severe timing errors due to the lack of water travel time trajectories to distribute corrections.
From a practical point of view, the QPE IRC-ICC corrections amount to space-time bias
correction. The improved QPE data can be used to build general QPE error prediction models such
as Liao and Barros (2023) and therefore correct remote-sensing products to improve orographic
QPE data to support hydroclimatic studies and model calibration under reduced forcing
uncertainty.

## 4. Discussion and Future Work

Limitations in this study stem mainly from computational constraints rather than the
methodology. A systematic definition of 24-hour flood duration is imposed,  implying that for
floods with longer high-flow tails slow contributions from deeper soil layers are not considered.
The current IRC-ICC framework was built to support flash flood studies and only utilizes shallow
subsurface moisture transport information, consistent with the governing role of shallow soil
moisture dynamics in steep topography. It is expected that for long duration precipitation events
or basins with large mild-slope areas, deeper interflows would play a much more important role in
improving flood timing, volume estimation and resulting QPE via IRC-ICC.
We plan to improve the StageIV-IRC product by further improving the IRC-ICC
framework through improved model physics and resolution and utilizing 3D velocity fields to
capture the full travel time distributions. When computational resources allow, the IRC can be
carried out at the same frequency as the model resolution, therefore eliminating any artifacts
produced due to inadequate sampling and updating of travel time distributions. This dataset can
even be used in near real time in operational hydrology to improve Quantitative Precipitation
Forecasts (QPF), advancing flood forecasting and emergency management.



### 5. Data Availability Statement


The StageIV-IRC dataset at 250 m 5 minute resolution for 26 basins and 215 events is
available at: https://doi.org/10.5281/zenodo.14028867.(Liao and Barros, 2024). Associated
geographic documentation of the selected basins is also provided via the same link.

### 6. Conclusion


QPE has been an enduring challenge particularly in complex terrain. Radar QPE are
plagued with uncertainties from multiple sources while rain gauge networks are scarce and suffer
from the lack of representativeness in the mountains. To address this grand challenge, we develop
a series of corrections from point-scale to watershed-scale encompassing event, climatology, and
water budget closure corrections for radar QPE: the IRC-ICC framework. To our knowledge, this
is the first QPE dataset aiming to close the water budget at high resolution for flood events,
consistent with fundamental physics at watershed scale,  and achieving superior hydrological
performance at sub-hourly scale in headwater basins.
The coupled IRC-ICC framework is applied to 26 headwater basins in the Appalachians
for 215 events with robust success yielding significant improvements in streamflow simulation,
particularly on flood timing and volume. The tracking algorithm in the IRC-ICC framework is
only updated when shifting from one hydrological window to another but not every time step. With
enough computational resources, post IRC-ICC QPE data should further improve by capturing
transient travel time distributions between model time steps.
Over 90% of the events have flood timing errors within one hour using the StageIV-IRC
product compared to fewer than 20% of the events without the use of IRC-ICC, while the median
KGE improved from 0.34 to 0.86 across the events. Results show that initial conditions are more



important for less severe precipitation events, especially during the slow rising period of
hydrograph, which influence subsequent streamflow simulations. It is also worth noting that
physical parameters used in this work are not calibrated for any precipitation event in any basin.
This physics-based IRC-ICC framework can capture the fundamental physics involved in flash
flood events, that is the fast hydrological response in surface and shallow subsurface soil layers
due to steep slopes and gravity, therefore skillful hydrologic prediction is achieved without model
calibration by instead focusing on getting the forcing right.

The IRC-ICC is a general framework that can be incorporated into any distributed

hydrological models. Thus, the StageIV-IRC dataset also enables meaningful intercomparison
among different radar QPE dataset, providing physics insights into QPE error structure from water
budget closure perspective, toward improving radar retrievals and to characterize radar specific
errors related to radar operations at high spatial resolution in the mountains. The demonstrated
success of StageIV-IRC in ungauged basins strongly supports the use of IRC-ICC in the vast area
of remote mountains worldwide where raingauges are generally not available. This dataset can be
utilized as a reference for building machine learning models (or even deep-learning models when
the number of studied precipitation events is expanded) that can learn the QPE uncertainties
conditional on time of day, weather, climate and geomorphological regimes for both radar QPE
analysis and forecasts, advancing the understanding of orographic precipitation uncertainties at
high resolution across global mountains.



**CReDiT AUTHOR STATEMENT**


M. Liao: Methodology, Data curation, Writing- first draft preparation, Analysis, Investigation. A.
P. Barros: Conceptualization, Methodology, Analysis, Writing- revisions and editing, Supervision,
Project administration, Funding acquisition.

**COMPETING INTERESTS**


The authors declare there are no competing interests.

**ACKNOWLEDGMENTS**


The work was supported in part by NASA Precipitation Measurement Mission Program under
NASA grant 80NSSC19K0685 with the second author, a NASA Earth System Science Fellowship
with the first author, and grant from the IBM Accelerator program with the second author.





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





## LIST OF TABLES

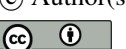



**Table 1** – Index, and coordinates for the raingauge stations marked in Figure1. The index is used
to identify specific gauges in some of the graphs. Two raingauges at Purchase Knob, a supersite
in the inner mountain region, are highlighted in bold font. Shaded rows indicate stations with
collocated raingauges that have different temporal resolution (e.g. tip size).

| NO. | Site ID. | Lat. | Lon. | Elev. (m) |
|---|---|---|---|---|
| 1 | RG001 | 35.39830 | -82.91300 | 1156 |
| 2 | RG002 | 35.41750 | -82.97140 | 1731 |
| 3 | RG003 | 35.38460 | -82.91610 | 1609 |
| 4 | RG004 | 35.36830 | -82.99020 | 1922 |
| 5 | RG005 | 35.40890 | -82.96460 | 1520 |
| 6 | RG008 | 35.38210 | -82.97360 | 1737 |
| 7 | RG010 | 35.45640 | -82.94680 | 1478 |
| 8 | RG100 | 35.58610 | -83.07250 | 1495 |
| 9 | RG100T | 35.58767 | -83.06468 | 1485 |
| 10 | RG101 | 35.57500 | -83.08820 | 1520 |
| 11 | RG102 | 35.56370 | -83.10360 | 1635 |
| 12 | RG103 | 35.55340 | -83.11790 | 1688 |
| 13 | RG104 | 35.55490 | -83.08800 | 1584 |
| 14 | RG106 | 35.43210 | -83.02910 | 1210 |
| 15 | RG109 | 35.49560 | -83.04040 | 1500 |
| 16 | RG110 | 35.54810 | -83.14820 | 1563 |
| 17 | RG300 | 35.72653 | -83.21692 | 1558 |
| 18 | RG301 | 35.70552 | -83.25595 | 2003 |
| 19 | RG302 | 35.72135 | -83.24675 | 1860 |
| **20** | **RG303PK** | **35.58610** | **-83.07253** | **1495** |
| **21** | **RG303S** | **35.76295** | **-83.16222** | **1490** |
| 22 | RG304 | 35.67010 | -83.18287 | 1820 |
| 23 | RG305 | 35.69150 | -83.13190 | 1630 |
| 24 | RG306 | 35.74597 | -83.17148 | 1536 |
| 25 | RG307 | 35.65163 | -83.19952 | 1624 |
| 26 | RG308 | 35.73027 | -83.18237 | 1471 |
| 27 | RG309 | 35.68297 | -83.15003 | 1604 |
| 28 | RG310 | 35.70273 | -83.12263 | 1756 |
| 29 | RG311 | 35.76507 | -83.14042 | 1036 |
| 30 | RG400 | 35.70273 | -83.12263 | 1756 |
| 31 | RG401 | 35.65163 | -83.19952 | 1624 |
| 32 | RG402 | 35.72135 | -83.24675 | 1860 |
| 33 | RG403 | 35.51777 | -83.10113 | 925 |
| 34 | RG407 | 35.51777 | -83.10113 | 925 |







**Table 2**: Hydrologic skill metrics used in this study.

| Metric | Description/Unit | Formula/Reference |
|--------|-----------------|-------------------|
| KGE | Kling-Gupta efficiency | Eq. (19) /Gupta et al. (2009) |
| NSE | Nash-Sutcliffe efficiency | Eq. (20) /Nash and Sutcliffe. (1970) |
| EV | Error in area under the hydrograph | Eq. (21) |
| EPT | Error in Time to Peak (minutes) | Time diff. between mid-points of rising limbs |
| EPV | Relative Error in Peak Volume | Eq. (22) |




**Table 3** - Basin information including basin index used in this work for simplicity, USGS
streamflow gauge ID, the corresponding drainage area, highest elevation in the basin and basin
relief.

| Basin index | USGS Gauge ID | Drainage area (km²) | Basin highest elevation (m) | Basin relief (m) | Location |
|---|---|---|---|---|---|
| 1 | 3544970 | 118.7 | 1442 | 847 | GA |
| 2 | 2178400 | 176.1 | 1629 | 1051 | GA |
| 3 | 3504000 | 149.9 | 1667 | 1032 | NC |
| 4 | 3497300 | 317.6 | 1999 | 1651 | TN |
| 5 | 3460000 | 148.1 | 1879 | 1174 | NC |
| 6 | 3456500 | 152.8 | 1873 | 1157 | NC |
| 8 | 344894205 | 41.3 | 1995 | 1221 | NC |
| 9 | 3463300 | 134.3 | 1989 | 1425 | NC |
| 10 | 3400500 | 234.7 | 1257 | 1257 | KY |
| 11 | 3479000 | 283.3 | 1772 | 1216 | NC |
| 13 | 3182700 | 447.3 | 1111 | 717 | WV |
| 14 | 2011460 | 194.4 | 1388 | 763 | VA |
| 15 | 1620500 | 54.5 | 1321 | 712 | VA |
| 16 | 3180500 | 426.8 | 1416 | 621 | WV |
| 17 | 3068800 | 437.1 | 1471 | 908 | WV |
| 18 | 1595000 | 234.8 | 1230 | 560 | MD |
| 19 | 1595300 | 130.3 | 1069 | 712 | WV |
| 20 | 1544500 | 445.9 | 765 | 457 | PA |
| 21 | 1422747 | 81.4 | 766 | 394 | NY |
| 22 | 1415000 | 106.8 | 1019 | 636 | NY |
| 23 | 1413398 | 152.8 | 1094 | 754 | NY |
| 24 | 13621955 | 41.7 | 1074 | 717 | NY |
| 25 | 1421610 | 51.3 | 970 | 497 | NY |
| 26 | 1074520 | 389.4 | 1582 | 1582 | NH |
| 27 | 10642505 | 294.9 | 1895 | 1693 | NH |
| 28 | 1137500 | 300.3 | 1894 | 1546 | NH |
| 29 | 1133000 | 183.2 | 975 | 719 | VT |
| 30 | 1055000 | 334.1 | 1143 | 975 | MAINE |









year reference period; c) Diurnal cycle of RMSE at hourly time-scale and seasonal-scale RMSE conditional on observed rainfall rate. Note only the winter season (JFM, January February, March) is demonstrated in this Figure.

**Figure 8** – The application of the coupled IRC-ICC framework to Basin05 (Cataloochee Creek Basin, NC) for 2017-10-23 event. This extreme hydrological response is caused by the remnants of Hurricane Nate in 2017. This figure includes **a)** the original simulation results using $STIV_{DBKC}$ as inputs; **b)** the dashed rectangular plot consisting of intermediate results including each iteration from the coupled IRC-ICC framework as explained in Figure 3; **c)** the equilibrium state reached by the coupled framework after 5 iterations, and **d)** KGE value compilation graph calculated using 15 minutes intervals for each iteration in the coupled framework.

**Figure 9** – The systematic application of the coupled IRC-ICC framework to the 28 basins selected in the Appalachians. The results include **a)** 5 events from the Southern Appalachians; **b)** 5 events from the Central Appalachians; and **c)** 5 events from the Northern Appalachians. The IRC-ICC KGE evolution plots from iterations are included below the hydrographs. The black dash line uses the original $STIV_D$ and the pink line is the IRC-ICC equilibrium state ($STIV_D^{IRC*}$), and the corresponding colored numbers are KGE values calculated at 15 minutes interval.

**Figure 10** – Event total precipitation maps for **a)** cold season events and **b)** warm season events. Each category includes 5 columns representing different events and 3 rows with the first row (**a1**, and **b1**) representing original precipitation input $STIV_{DBKC}$, and the second row (**a2**, and **b2**) representing $STIV_{DBKC}^{IRC*}$ from IRC-only framework, and the third row (**a3**, and **b3**) representing $STIV_{DBKC}^{IRC*}$ from the coupled IRC-ICC framework.

**Figure 11** – Summary charts of precipitation statistics for all event-total precipitation maps. Basin mean and standard deviation for each event are represented by circles and triangles in the top and bottom panel, respectively. Each panel is separated into 3 sub-regions by vertical black lines: the Southern Appalachian Mountains, Central Appalachian Mountains, and Northern Appalachian Mountains (SAM, CAM and NAM). The list of events in Basin 05 (with event number ranging from 55 to 108) in the SAM is highlighted by a green rectangle for further discussion in the text. The average values of all events for both the mean and the standard deviation are calculated and shown in the top right corner. Black color and pink color represent pre and post IRC-ICC QPE statistics, respectively.

**Figure 12** – Summary charts of hydrologic skill metrics for all events. Horizontal green dash lines (i.e. the perfect situation) and green envelopes are for reference purposes. Hydrologic statistics are explained: EPV: Error in Peak Volume (Unit: %), EPT: Error in Peak Timing (Unit: minutes), EV: Error in flow Volume (Unit: %), note KGE is calculated using **15-minute** intervals over a 24 hour period. Pink dots and black dots represent post IRC-ICC results, and original inputs results, respectively (each dot represents one event). Each panel is separated into 3 sub-regions: the SAM, CAM and NAM. Histograms graphs are attached right next to the scatter plot.

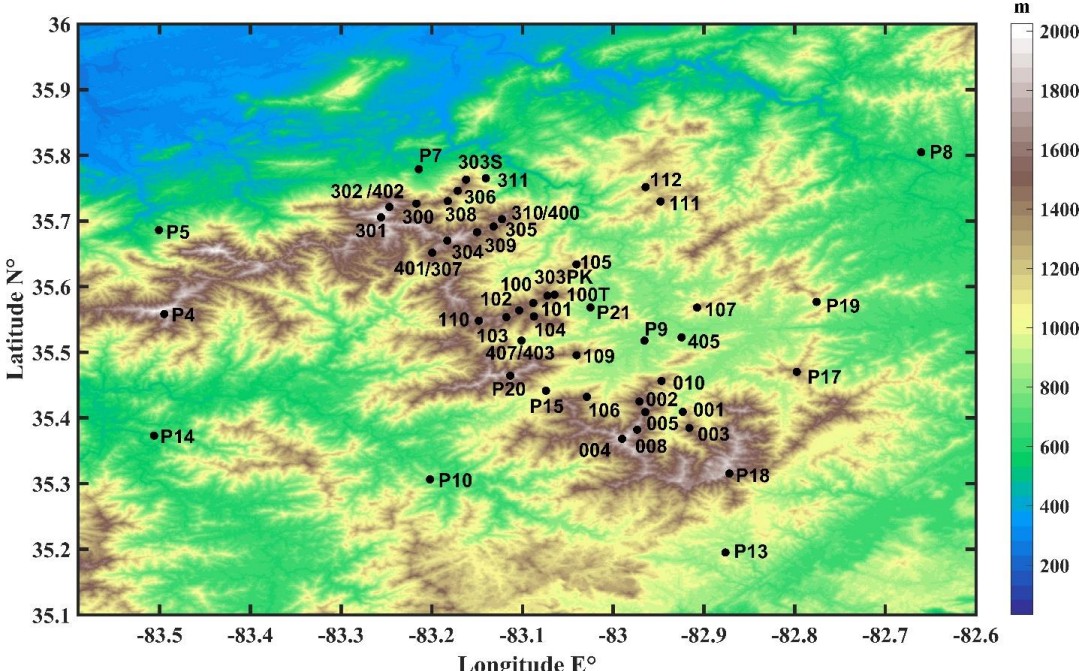

**Figure 1 -** Map of ground-based observations. Locations marked by numbers-only are raingauges; locations marked by numbers preceded by P are disdrometers. See Table 1 for list of stations and geographical coordinates.



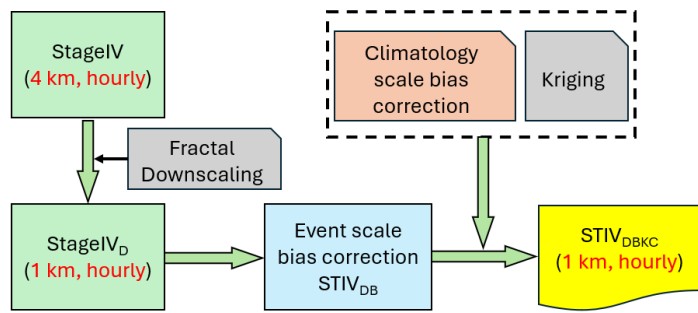

**Figure 2** – Workflow to generate the product STIV$_{DBKC}$.

1523

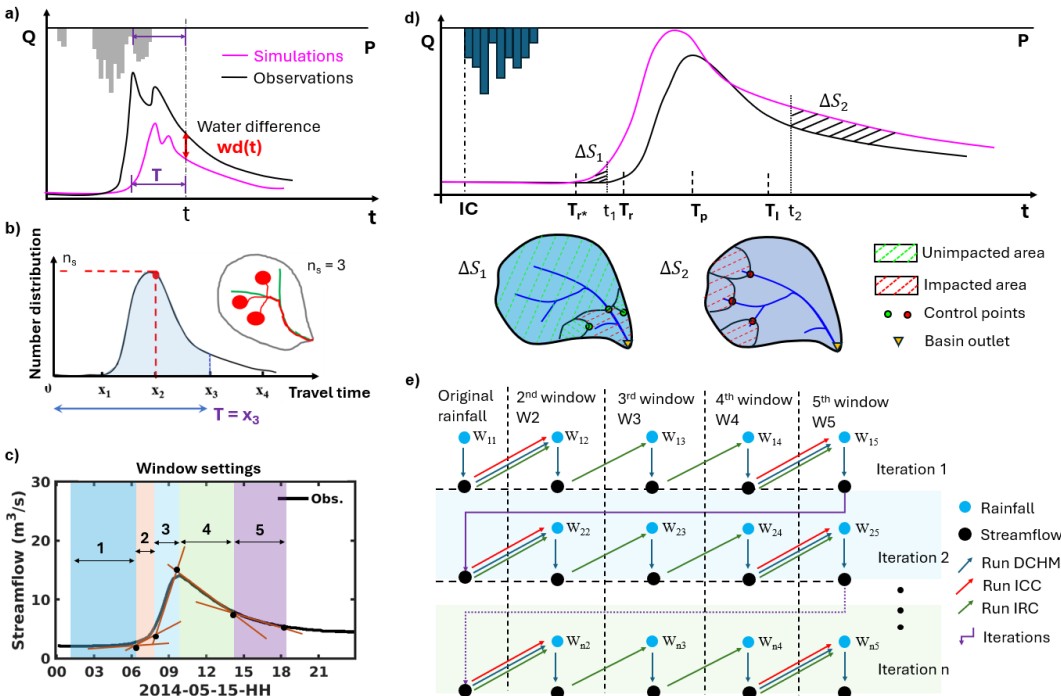

1524

**Figure 3** – An illustration of the structure of IRC, ICC and the coupled IRC-ICC framework including **a)** the residual hydrograph between the observed and simulated discharge, with the discharge water difference *wd(t)* being distributed across the time window T; **b)** Example of travel time distribution TT(t) and map (inset) illustrating a hypothetical distribution of runoff source areas (in red, ns=3) with travel time $x_2$ contributing to streamflow at time t, meaning that at time $t-x_2$ there are three pixels (ns=3) generating runoff that reaches the outlet at time t. T is the time window over which runoff source areas with TT < T are mapped and the inverse rainfall correction (IRC) are applied; **c)** Example of IRC windows guided by timescales of dominant hydrological processes. The first window solely covers the initial streamflow conditions before the target event. The second window depicts the early rising limb of the hydrograph. The third window captures the steep rising limb of the hydrograph until it reaches the peak flow. The fourth and fifth windows correspond to interflow-dominant and baseflow-dominant stages of the recession curve respectively, separated by the recession inflection point; **d)** A schematic drawing that illustrates the Initial Condition Correction (ICC). Characteristic timing of the hydrographs: $T_{r*}$ and $T_r$ are rising points of simulated and observed hydrograph respectively. $T_p$ is peak timing of observed flow. $T_I$ is the inflection point of observed flow. Flow differences between simulated and observed flow at $t_1$ and $t_2$ are denoted as $\Delta S_1$ and $\Delta S_2$ respectively as examples for discussion in the text. IC, P, and Q stand for initial condition, precipitation and discharge, respectively; **e)** A recursive framework consists of Initial Condition Correction and Inverse Rainfall Correction (i.e. the coupled IRC-ICC framework) for illustration purposes.

1545

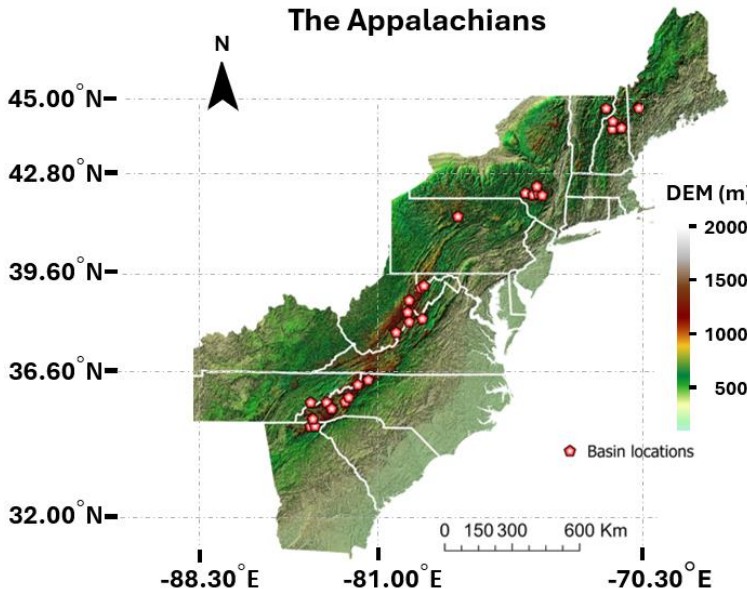

**Figure 4** – Headwater basins selected in the Appalachian Mountains, USA. The USGS gauge ID for each basin and basic information are listed in Table 2. For simplicity, sub-regions are identified and conveniently named as: Southern Appalachian Mountains (SAM, including Basin 01-11), Central Appalachian Mountains (CAM, including Basin 13-20), and Northern Appalachian Mountains (NAM, including Basin 21-30).

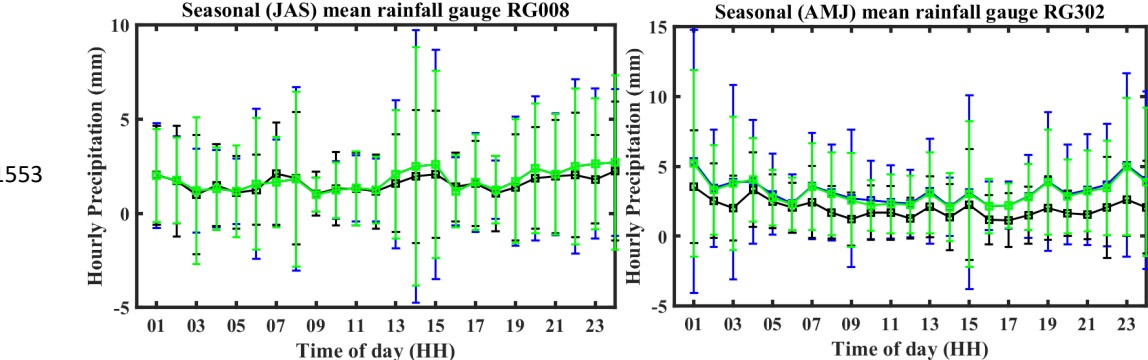

**Figure 5** - Diurnal cycle of rainfall (mean and ±standard deviation) for different seasons and gauge locations.  Left panel - Summer (JAS: July-August-September) at RG008 in the eastern ridges. Right panel – Spring (AMJ; April-May-June) at RG302 in the western ridges. Raingauge measurements (blue); StageIV$_{DBK}$ (black); StageIV$_{DBKC}$ (green).

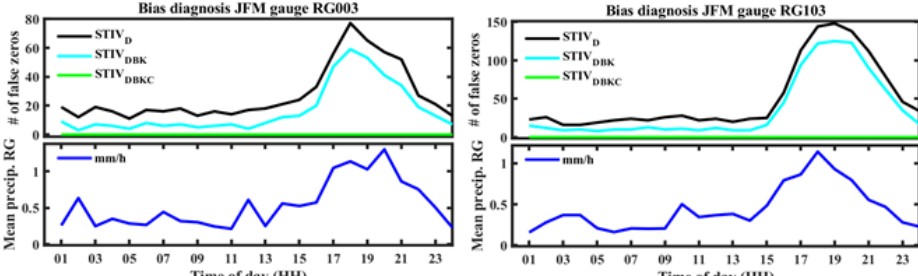

**Figure 6** –Top row - Wintertime (January-February-March, JFM) diurnal cycle of missing precipitation in the eastern ridges (RG003) and in the inner region (RG103) for each of the RR products: . Bottom row- same as top row for the raingauge climatology of hourly rainfall (blue). StageIV$_D$ (black); StageIV$_{DBK}$ (cyan); StageIV$_{DBKC}$ (green).





1565

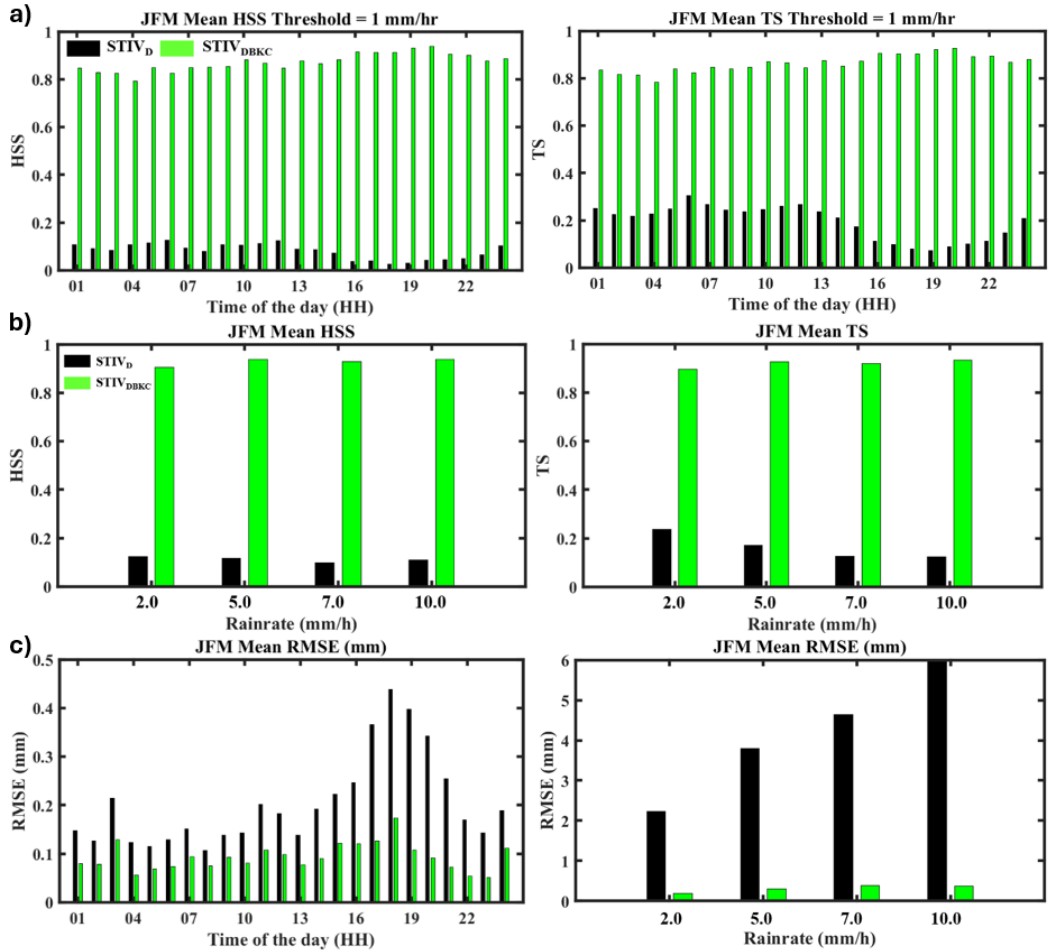

1566

**Figure 7** – Statistics summary: a) Diurnal cycle of mean HSS and TS statistics including all
raingauges over the 10-year reference period (2008-2017): STIV$_D$ (black) and STIV$_{DBKC}$ (green);
b) Seasonal mean HSS and TS statistics conditional on different rainfall thresholds over the 10-
year reference period; c) Diurnal cycle of RMSE at hourly time-scale and seasonal-scale RMSE
conditional on observed rainfall rate. Note only the winter season (JFM, January February, March)
is demonstrated in this Figure.

1573

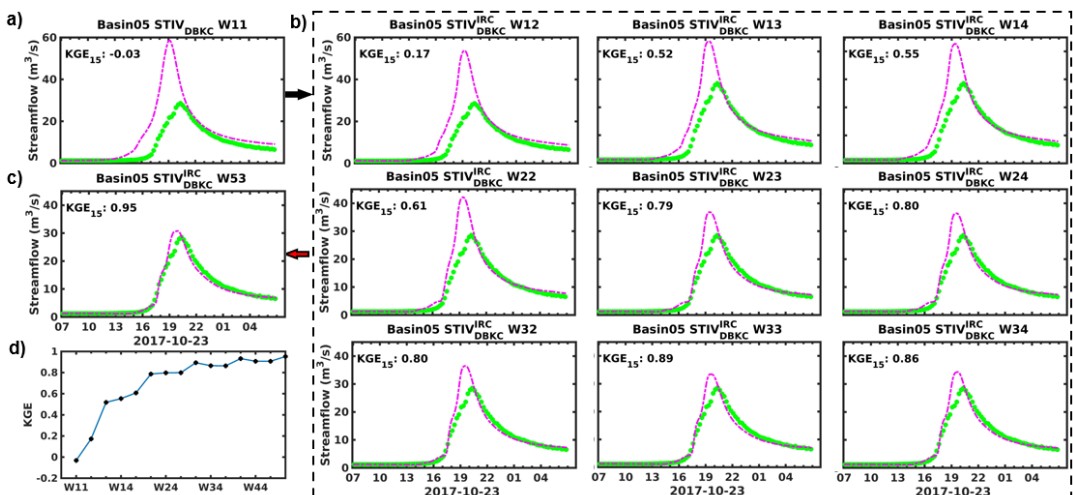

**Figure 8** – The application of the coupled IRC-ICC framework to Basin05 (Cataloochee Creek Basin, NC) for 2017-10-23 event. This extreme hydrological response is caused by the remnants of Hurricane Nate in 2017. This figure includes **a)** the original simulation results using $STIV_{DBKC}$ as inputs; **b)** the dashed rectangular plot consisting of intermediate results including each iteration from the coupled IRC-ICC framework as explained in Figure 3; **c)** the equilibrium state reached by the coupled framework after 5 iterations, and **d)** KGE value compilation graph calculated using 15 minutes intervals for each iteration in the coupled framework.

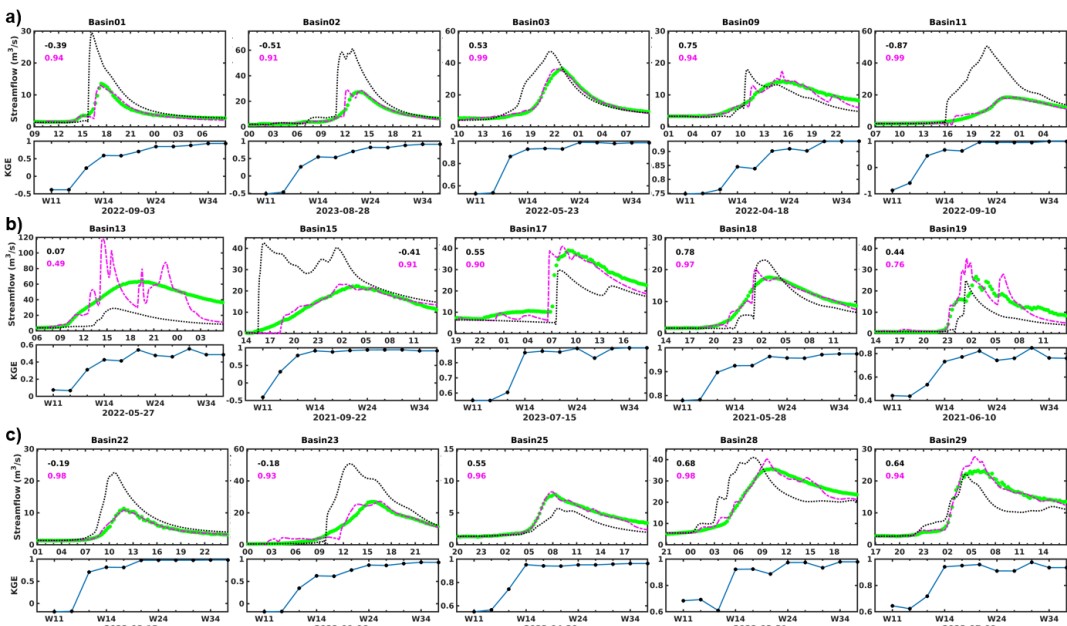

**Figure 9** – The systematic application of the coupled IRC-ICC framework to the 28 basins selected in the Appalachians. The results include **a)** 5 events from the Southern Appalachians; **b)** 5 events from the Central Appalachians; and **c)** 5 events from the Northern Appalachians. The IRC-ICC KGE evolution plots from iterations are included below the hydrographs. The black dash line uses the original $STIV_D$ and the pink line is the IRC-ICC equilibrium state ($STIV_D^{IRC*}$), and the corresponding colored numbers are KGE values calculated at 15 minutes interval.

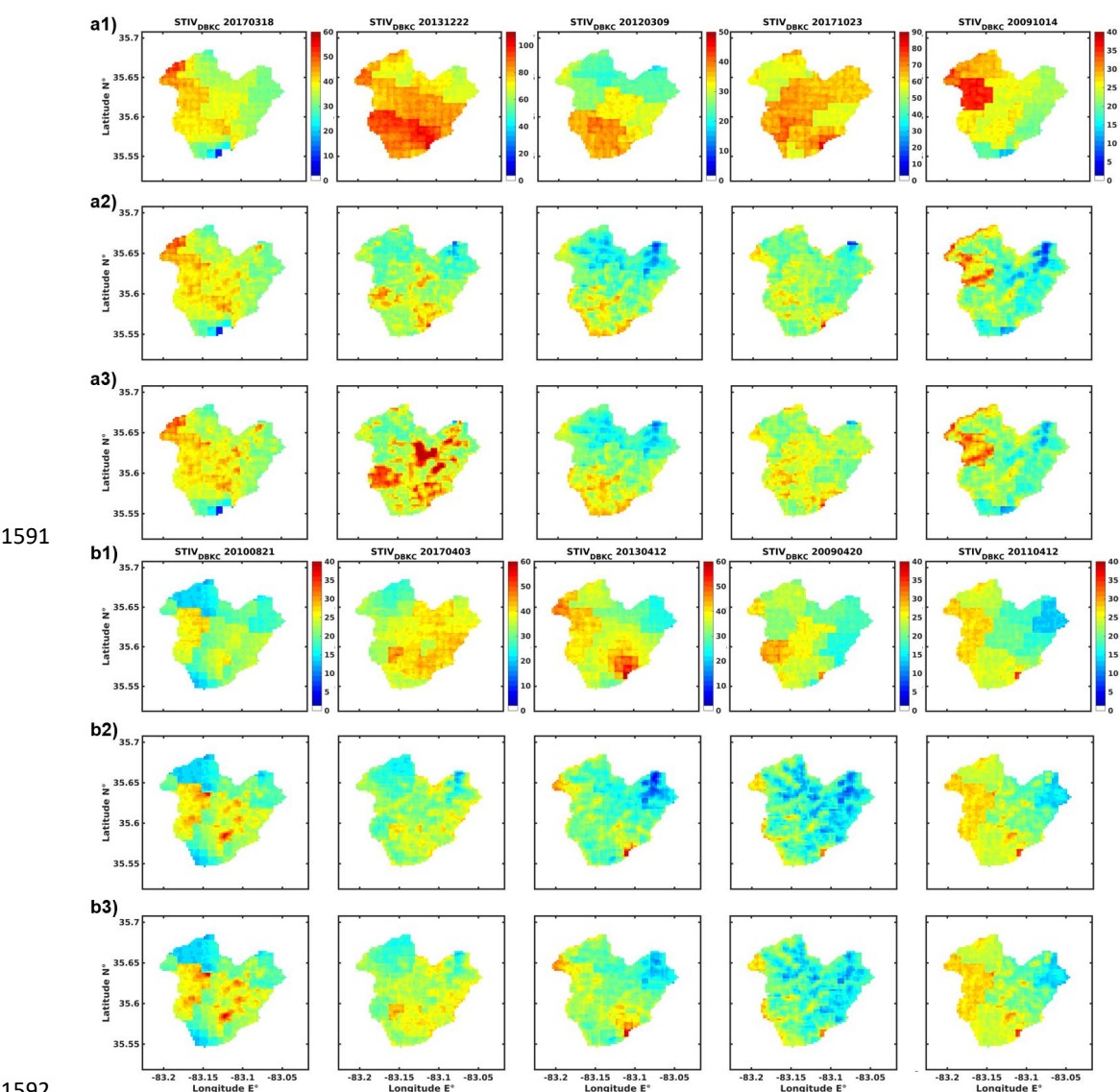


**Figure 10** – Event total precipitation maps for **a)** cold season events and **b)** warm season events.
Each category includes 5 columns representing different events and 3 rows with the first row (**a1**,
and **b1**) representing original precipitation input $STIV_{DBKC}$, and the second row (**a2**, and **b2**)
representing $STIV_{DBKC}^{IRC*}$ from IRC-only framework, and the third row (**a3**, and **b3**) representing
$STIV_{DBKC}^{IRC*}$ from the coupled IRC-ICC framework.



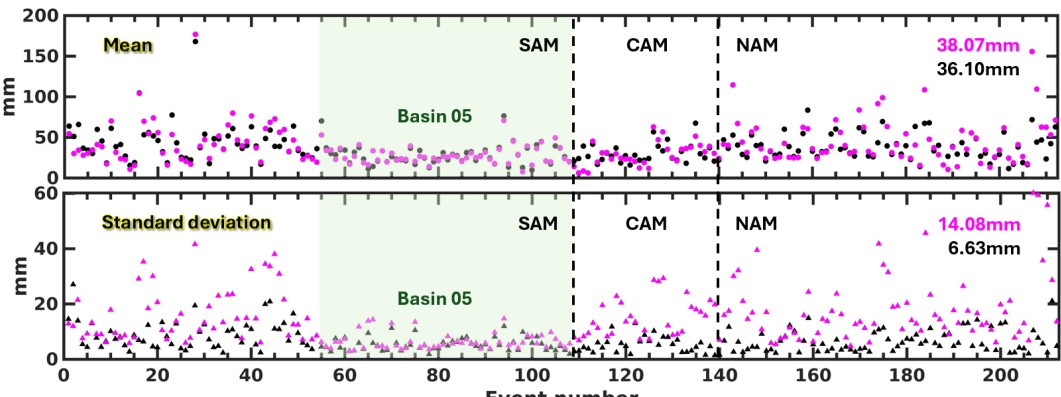


**Figure 11** – Summary charts of precipitation statistics for all event-total precipitation maps. Basin mean and standard deviation for each event are represented by circles and triangles in the top and bottom panel, respectively. Each panel is separated into 3 sub-regions by vertical black lines: the Southern Appalachian Mountains, Central Appalachian Mountains, and Northern Appalachian Mountains (SAM, CAM and NAM). The list of events in Basin 05 (with event number ranging from 55 to 108) in the SAM is highlighted by a green rectangle for further discussion in the text. The average values of all events for both the mean and the standard deviation are calculated and shown in the top right corner. Black color and pink color represent pre and post IRC-ICC QPE statistics, respectively.



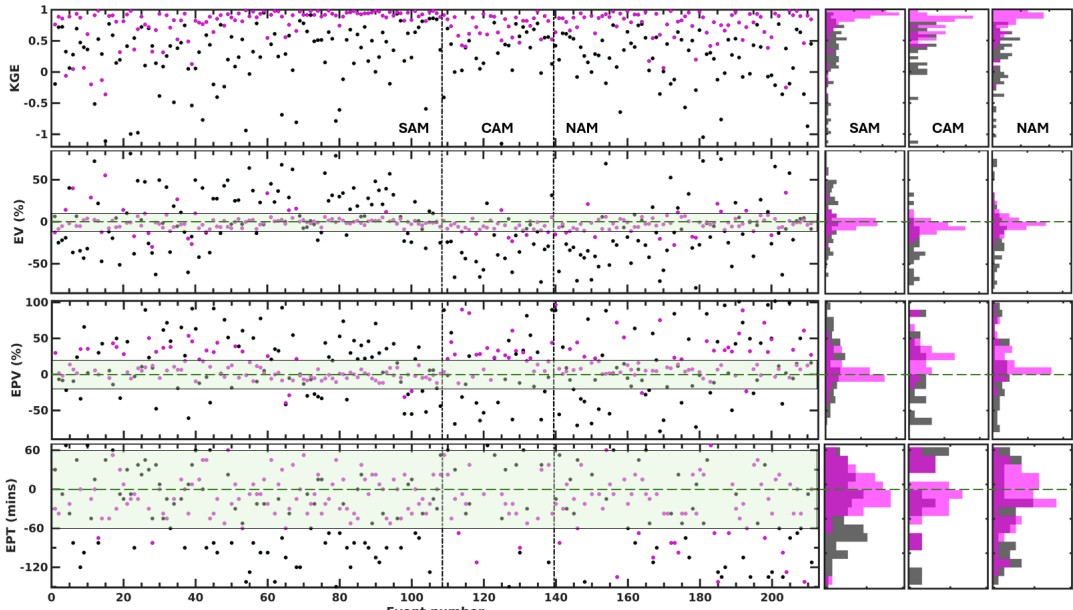


**Figure 12** – Summary charts of hydrologic skill metrics for all events. Horizontal green dash lines
(i.e. the perfect situation) and green envelopes are for reference purposes. Hydrologic statistics are
explained: EPV: Error in Peak Volume (Unit: %), EPT: Error in Peak Timing (Unit: minutes), EV:
Error in flow Volume (Unit: %), note KGE is calculated using **15-minute** intervals over a 24 hour
period. Pink dots and black dots represent post IRC-ICC results, and original inputs results,
respectively (each dot represents one event). Each panel is separated into 3 sub-regions: the SAM,
CAM and NAM. Histograms graphs are attached right next to the scatter plot.