# Peer review of "StageIV-IRC – A High-resolution Dataset of Extreme Orographic Quantitative Precipitation Estimates (QPE) Constrained to Water Budget Closure for Historical Floods in the Appalachian Mountains"

_Earth System Science Data, 2024_

## Author Comment (AC1)

RC1: 'Comment on essd-2024-513', Anonymous Referee #1, 30 Jan 2025

The authors present a QPE data for a large number of watersheds throughout the Appalachians developed using inverse method for correcting radar/gage QPE based on observed streamflow and a hydrologic model. I still start by disclosing that I also reviewed 2024WR038446 (Water Resources Research), which is among the as-yet unpublished studies by the same authors. As with that manuscript, I found this study to be intriguing and somewhat challenging, and ultimately have the same general concern as I expressed (not as clearly as here) in my review for that prior work.

Thank you for reviewing this manuscript. The replies are in blue fonts.

Basically, my interpretation is that the authors have developed an approach for adjusting QPE based on back-trajectories of simulated streamflow such that the water budget closes at the event scale. This makes sense; I toyed with similar ideas myself years ago (though never did any real work on it). It does raise a potential concern, that I don't feel the authors did a great job addressing in either study. Specifically, if you use this approach, it seems to me that the appropriate way of judging success is whether the adjusted rainfall looks "better," i.e. closer, in terms of amount and spatiotemporal pattern, to some reference precipitation. Obviously, that's hard to do, since we don't have good reference precipitation in many locations. Instead, the authors show that the simulated hydrographs using the corrected precipitation have improved. This doesn't seem convincing—of course they have improved. You've adjusted the rainfall specifically to make sure that the hydrographs improve; then used the improved hydrographs as evidence that you have fixed the rainfall problems. But does that mean that the rainfall is more accurate? If the hydrologic model is good (and I trust that the authors' model is good) then the answer is "probably." If the model is not good, the answer is "probably not." The authors don't really answer the question.

Thank you for pointing out issues regarding this approach (i.e. IRC) and the need to validate post-IRC QPE. As mentioned in your comment, many places don't have good reference precipitation data, which is particularly true for mountainous regions. In response to this comment, we downloaded the Multi-Radar/Multi-Sensor (MRMS) data, and conducted a comparison between post-IRC QPE and MRMS QPE. Note MRMS data suffer from a relatively low radar quality index (RQI) for radar gaps in the mountains. For comparison, basin-averaged event total QPE is calculated for each dataset (i.e. StageIV$_D$, StageIV$_D$IRC, MRMS) and for each basin and each event. The results are shown in Figure S1.

[Figure]

Figure S1 – Comparison of different QPE products for event total precipitation estimation (basin-averaged). Each point represents one event.

MRMS is often considered more advanced than StageIV in the scientific community because it incorporates much more data from various data sources, making it potentially more accurate in precipitation estimation than StageIV. Figure S1 shows that when post-IRC products yield high KGE values (>0.8, accounting for 52.5% of the events), StageIV$_D$ IRC has better agreement with MRMS.  When post-IRC event rainfall generates relatively bad hydrological simulation (with KGE<0.8), it is expected that StageIV$_D$ IRC and MRMS have a relatively large discrepancy in this scatter plot. Figure S1 also points out that MRMS generally agrees well with StageIV for intense rain rate that occurs in June, July and August. This information is also reported in other studies.

A closer look at the outliers of StageIV$_D$ IRC in Figure S1 indicates a dependency of IRC performance on basin size. Relatively larger basins (mostly >200 km$^2$) usually produce lower KGE values (<0.8), indicating the IRC is not as effective in larger basins as in smaller headwater basins. This is also illustrated in the manuscript because the current version of IRC only uses shallow-layer travel time distributions up to 24 hours. For larger basins, it is expected that slower hydrological response from deeper layers becomes increasingly

important simply because of larger areas of relatively flat floodplains. Water travel time distributions from deeper layers should be considered for larger basins (>200 km$^2$), and long-lasting precipitation events (>24 hours).

Besides comparison against MRMS data, the authors also investigated available raingauge data. However, only one raingauge at Mill Gap in Virginia from COOP v2 dataset is available. This raingauge is unfortunately located in Basin 14, which is one of the two basins studied (Basin 13 and 14) that have complicated subterranean structures (i.e. Karst terrain), where DCHM performs poorly due to the lack of a Karst terrain module. Therefore, the post-IRC QPE is not reliable in these two basins, and this is discussed in the manuscript, thus no comparison against raingauges is executed in this study. However, raingauge comparison is done in the very original method paper (Liao and Barros, 2022) using a network of raingauges at high elevations in the Cataloochee Creek Basin.

Furthermore, MRMS data are downscaled to 250m resolution using nearest neighbor interpolation for hydrological simulation. The histograms of KGE distributions for various QPE data products are demonstrated in Figure S2.

[Figure]

Figure S2 – Histograms of KGE values of studied events for various QPE products.

Figure S2 shows that the hydrological performances of StageIV$_D$ and MRMS$_D$ are not largely different, which is expected because they are using the same radars. The median KGE for both datasets are below 0.2. However, the median KGE for StageIV$_D$ IRC is above 0.8 for these extreme events.

I would appreciate the authors' response to that criticism. In addition, the authors need to state more clearly the differences between this study and others, particularly Liao and Barros (2024a), which I have reviewed, and Liao and Barros (2024b), which I have not. I guess this study is essentially a "scaling up" of the methods from those papers to more watersheds over a larger region? Fine, but please state it clearly.

Yes, this paper is a 'scaling up' of the methods from previous papers. Previous papers focus on a couple of headwater basins in the Southern Appalachians. This paper focuses on 30 headwater basins with over 200 extreme precipitation events in the entire Appalachians. Weather and climate regimes are dramatically different between Basin 01 (located in Gorgia) and Basin 30 (located in Maine). The headwater basin sizes range from 40km$^2$ to 450km$^2$. We also plan to submit another paper (Liao and Barros, 2025, in preparation) that includes 500 mountainous basins with over 10,000 extreme events including the Alps, the Andes, the Himalayas and the Brazilian Highlands, where the IRC method is demonstrated with significant success.

I will add that, similar to the earlier manuscript, the writing and overall presentation quality of figures should be improved. There are a large number of minor grammatical problems, especially with run-on sentences (three in the abstract alone) and missing articles (mainly "the") and some verb tense problems. These didn't make it impossible to understand the paper but do distract from the study's strengths. The figures should include legends, readable font sizes,

Thank you for the comment. We attached a revised manuscript along with the replies. Specifically, font size is increased in Figure 3. Legends are added for Figure 5-11. Excessive examples in Figure 8-10 are removed to accommodate font sizes.

The statement "uncertainty from the model and model parameters is assumed to be negligible" is not really reasonable, and is inconsistent with the following sentence that states that these have "secondary importance." "Secondary importance" is ok (at least for flood events, since as you note, forcing uncertainty will be large), "negligible" is not. Indeed, you aren't neglecting them in your method. Instead, you transfer the calibration effort from model parameters to rainfall. That is perhaps a reasonable thing to do for flood simulations with a high-quality distributed hydrologic model, as in this study. But it is an issue that should be more clearly acknowledged in your studies. And returning to my first concern, it is problematic if you are unable to quantify whether that produces improved rainfall—if your rainfall is practically incorporating model structure and parameter error,

there is the risk that it produces unrealistic rainfall outcomes. But in your study, we are left to wonder.

Thank you for improving the rigor of this article. The authors acknowledge that parameter uncertainty and model uncertainty can have a large impact on model performance, especially for simulating the hydrological response of moderate and less significant precipitation events. Therefore, the study samples of this study only include rainfall that produces flood peaks greater than 95 percentiles of streamflow measurements, where precipitation uncertainty should dominate over other sources of uncertainties.

I realize that I might be misunderstanding the issue entirely. If so, please clarify.

Thank you again for reviewing this manuscript. The replies above should address the issues raised in the reviews.

---

## Author Comment (AC2)

We thank the Reviewer for the comments. Our detailed replies are in blue below.

RC2: 'Comment on essd-2024-513', Anonymous Referee #2, 27 Feb 2025

The authors developed a framework based on their previous efforts on inverse rainfall-runoff modeling, and bias correction methods. Based on the framework, they developed a bias-corrected rainfall product that consists of 200+ storm events spreading across 20 basins in the Appalachian Mountains. The new rainfall product maintains a better basin-scale water budget and shows promising results for flash flood modeling in this region. As a manuscript submitted to ESSD, the uniqueness, usefulness, and completeness of the dataset should the most important criteria. While I have no concerns about the uniqueness or completeness, the usefulness of the bias-corrected dataset is questionable. In addition, the dataset covers only a very limited region, only head watersheds in the Appalachians. Whether the proposed framework to other watersheds or storm events is another concern of mine. This is because there are many subjective choices in the framework. Justification of these routines is needed.

The first figure below is from Saharia et al. (2017) and it shows the alignment of flash flood climatology with the Appalachian Mountains east of 105° in the US. A recent example is Hurricane Helene in September 2024 which caused over 200 deaths and $50 billion in property damage in the Southeast U.S. The second figure is a distribution of global flood hazards from the World Bank and Columbia University that illustrates the alignment of the most severe flooding with topographic features globally. Approximately 2 billion people live in such regions around the world, and apart from massive earthquakes, flashfloods are the deadliest natural hazard due to the combination of severity and frequency.

[Figure]

FIG. 2. Distribution of observed flashiness (0–1) over CONUS. The bounding boxes highlight known flash flood hotspots: 1) West Coast, 2) Arizona, 3) Front Range, 4) Flash Flood Alley, 5) Missouri Valley, and 6) the Appalachians.

From Saharia et al. 2017

[Figure]

As shown by Barros (2013, see Fig. 19a below), the number of gauged basins in mountainous regions is very small globally and strongly decreases with elevation. Whereas the actual number changes in any given year, the situation has not improved in the last ten years, and indeed the number of well-maintained stream gauges that can be used reliably to evaluate and calibrate hydrological models and hydrological forecasts is even smaller and has decreased due to impacts of geopolitical instability and financial woes of regional agencies. The number of precipitation gauges is much higher but its density also strongly decreases with elevation. Studies of the historical record of exceptional floods (see figure below from Coast and Jarrett, 2008) show the steepest increase of flood magnitude with drainage area between 1 km$^2$ and 250 km$^2$, the typical range of headwater basins in complex terrain. It is therefore critical that high quality data sets be available to support hydrologic (or AI based) model evaluation and calibration. Given the foundational importance of precipitation, we argue that having access to this high-quality data set is of great value and benefit to the community. This is the reason why we are making it available.

[Figure]

From Barros (2013).

[Figure]

Adapted from Costa and Jarrett (2008).

The Reviewer's point that there are approximations in the implementation of the IRC framework is well taken. These are well documented and justified, and the skill of the hydrologic metrics is very high. In the future, we will plan to rerun the full framework using more powerful computational resources; thus, we refer to the current data as version 1 (e.g., StageIV-IRC.1) . These do not limit the utility of the dataset however. There are thousands of papers published in the peer-reviewed literature reporting on systematic evaluation of various precipitation products including optimal combinations of satellite, ground-based radar and rain gauges. None of such products meets the requirements needed at the small spatial scales of headwater basins, let alone capturing the high space-time variability of precipitation. Because of the readily available data sets for the continental US including streamflow and precipitation, and the quality of streamflow observations at gauges maintained by the USGS, the selected basins across the Appalachian Mountains provide a unique opportunity to generate a high-quality precipitation product that includes significantly different hydroclimatic regimes and physiographic settings suitable for research.

This is the first time that the previously established inverse precipitation correction approach (i.e. IRC) is systematically applied to a wide range of basins with drainage areas

ranging from 40km$^2$ to 500km$^2$. Note Basin01 and Basin30 in this study are over 2,000 kilometers apart with drastically different weather regimes and topography. Another reason that this area is chosen is because these studied headwater basins are equipped with USGS streamgauges with 15-minute monitoring resolution, which allows us to study flash flood events in this region as rainfall runoff response can be as fast as 30 to 90 minutes in steep terrain. Similarly, the reasons for not studying larger basins (>1000 km$^2$) are: 1: rainfall runoff response in bigger basins is relatively slower, therefore not meeting the criterion of flash floods timescales (<6 hours). 2: hydrological model resolution in this study is high (250m, 5 minutes) to capture flash floods, which makes it less suitable to run the same model configurations for bigger basins due to computational constraints. The authors also plan to use coarser resolutions and apply the same methods to larger basins in the future to study other flood events that last over 12 to 48 hours.

I agree that the golden rule is to test the performance of the dataset through hydrological modeling, but this does not mean that we should "force" it to happen. The core of the proposed bias-correction framework has been developed in the authors' previous studies. I would thus suggest the authors to pick up either one route (framework or dataset) and resubmit to a more suitable journal. I have some other concerns, which are listed below (not necessarily in the order of importance).

We respectfully disagree, and this follows from the arguments stated above.  The point of this manuscript is to provide this unique data set that was developed using the IRC framework along with detailed documentation how the data were obtained.  This is not different from the publications that describe products like the ERA 5, or GPCP, or IMERG, and many others.  Such products used to drive hydrologic models and for model calibration are "forced" to agree with rain gauge or radar observations in some statistical sense without meeting fundamental hydrologic criteria such as water budget closure besides exhibiting very large biases and lacking the sub-hourly space-time complexity of realistic storms. Whereas no doubt further improvements and bigger datasets will become available in the future, it is time to start a new generation of hydrology-centered precipitation products.

1. **Model uncertainty**. The authors emphasize that they are use a non-calibrated model in this region and model uncertainty is minimal. However, they attribute some of the poor performance in the new product due to model uncertainty (not capable of groundwater modeling). This is problematic, especially when transferring the framework to other regions with diverse land surface properties of runoff-generation mechanisms.

We appreciate the Reviewer's comment regarding model uncertainty, and we respectfully disagree with the statement above that is not accurate. The DCHM is capable of groundwater modeling. This model has been extensively used for the last 25 years in the Appalachians with parameters and parameterizations strongly tied to basin physics. What the model does not do is to simulate the complex subterranean hydraulics in karst terrain. Indeed, we are not aware of models of the same family as DCHM that do so. This is a question of model structural uncertainty as discussed in the manuscript. Consequently, the IRC precipitation for events in these basins are not expected to do well. We could have pretended that we did not select any basins in karst terrain. However, that would ignore an important element of the diversity of physiographic settings that surely can be found in other mountainous regions around the world as stated by the Reviewer. Therefore, it is critical that the challenges of karst hydrology and hydraulics be highlighted. This is like flagging a remote sensing product like for example SMAP soil moisture in the Amazon due to the limitations of soil moisture retrieval in densely forested areas.

This matter has no implications for transferability. In this paper we are making available a data set of extreme precipitation events that caused flash-floods for specific watersheds. We provide exhaustive description and documentation of how the data were obtained. ESSD is a journal with a well-defined mission to published data sets. This is why we submitted here.

**Presentation quality.** I would suggest the authors to substantially improve the presentation quality if resubmitted to other journals. This includes the structure of sentences which are too complicated to be understood, clear structure of the Introduction, more details for the IRC framework (right now it is only described in the figure), and less details in the equation and metrics of model evaluation. In addition, the manuscript lacks a map that clearly show the study region, including the watersheds, rain gauges, and different regions. This makes readers outside of US a miserable experience.

More details of the IRC are included in the revised manuscript along with new Figure A1-A3 in the Appendix. A new map showing the studied basins is included, and an improved version of Figure 4 that shows the regional perspective is added. In addition, links to the sites where all data are publicly available were added throughout. We regret causing frustration with the lack of these details. The manuscript was revised for clarity.

**Introduction.** The Introduction session should be reorganized in a more concise and logical way. The currently broad theme matter distracts from a focus on flash floods in mountainous areas. The second paragraph offers irrelevant context to the overall topic. It is vital to emphasize the significance of high-resolution precipitation estimates specifically in these regions. Instead of laboring extensively over the methods and details of this study, try to present a comprehensive overview of possible solutions to the challenges in QPE. The importance of Appalachian region should be highlighted as well. Why do the authors believe the new product should be able to contribute to earth system science?

Thank you for your suggestions for the introduction. The second paragraph is revised with a focus on flash floods. The importance of Appalachian region is explained according to the previous comment and is included in the revised manuscript. The importance of this work is to illustrate the uncertainties involved in the widely used radar QPE dataset (i.e. StageIV) and provide an improved dataset that can close the water budget at basin scale for flash floods.

2. More specific comments. I will not elaborate them all. They can wait till later rounds of reviews if applicable.

- Line 534-543: This paragraph is not appropriate in the Results session.

  Thank you for your suggestions. This paragraph is therefore eliminated from the results session.

- Line 603-605: A lack of understanding in Karst-terrain physics does not justify the omission of model parameter calibration.

  This sentence is meant to point out that model parameter calibration can hinder the understanding of physics and hide/compensate for errors in precipitation. This sentence is revised for clarity.

- Line 632-644: Discuss whether all flood-generating storms in mountainous regions align with terrain gradients and whether the greater consistency between precipitation spatial pattern and terrain gradients indicates better bias correction.

  The spatial co-organization of orographic precipitation and topography across all mountain ranges (e.g. Konrad II, 1994; Smith et al., 2011; Forestia and Pozdnoukhov, 2012; Wolvin et al., 2024) is well documented in the literature including several publications on precipitation processes in the Southern Appalachians. In the Southern Appalachians in particular this is the case for heavy precipitation events tied to convective activity. A statement to this effect was added to the manuscript and references were included.

In addition, to further evaluate the effectiveness of bias correction, a comparison between StageIV and StageIV-IRC against MRMS, a radar-based product like StageIV (https://www.nssl.noaa.gov/projects/mrms/) using the same radar network but at higher spatial resolution (1 km) and different raingauge corrections, was conducted. MRMS is widely considered as the most advanced QPE data in the US, though large precipitation uncertainty still exists in the mountains due to low radar quality index and wide radar gaps. Note that there is no "true" reference product due to the lack of raingauges, and thus the comparison here first shows that StageIV and MRMS are well aligned as expected (see also Gao et al. 2021) while there is significantly more variance for StageIV-IRC with better hydrologic simulations.

Figure S1 shows overall good agreement between StageIV-IRC and MRMS QPE data, while StageIV-IRC produces much better hydrological responses as demonstrated in the manuscript. DEM maps are plotted as additional Figure A2 in the Appendix.

[Figure]

Figure S1 – Comparison of different QPE products for event total precipitation estimation (basin-averaged). Each point represents one event.

- Figure 5-7: Explain why results for specific seasons are shown.

In this region, there are two primary precipitation regimes: long-lasting stratiform precipitation from shallow precipitation systems in the cold season (January, February and March), and high intensity thunderstorms in the warm season (July, August and September). Figure 5 shows the warm season and Figure 6 shows the cold season. A common issue is the missing detection of shallow precipitation systems

during late afternoon in the winter in this region which is highlighted in Figure 6. Figure 7 shows all metrics for the same season as in Figure 6.

- Figure 11: Show the terrain gradients, which makes it comparable to the spatial pattern of precipitation.

  Based on previous comments, terrain gradients are plotted in the Appendix Figure A2 using the example of Basin 05. The slope map of Basin 05 matches well with rainfall corrections in Figure 10 using event 2012-03-09 and event 2009-10-14 as examples, and the figures are demonstrated below (event-total rainfall are plotted):

[Figure]

Reference:

Konrad II, C. E. (1994). Moisture trajectories associated with heavy rainfall in the Appalachian region of the United States. *Physical Geography*, *15*(3), 227-248. https://doi.org/10.1080/02723646.1994.10642514

Smith, J. A., Baeck, M. L., Ntelekos, A. A., Villarini, G., & Steiner, M. (2011). Extreme rainfall and flooding from orographic thunderstorms in the central Appalachians. *Water Resources Research*, *47*(4). https://doi.org/10.1029/2010WR010190

Foresti, L., & Pozdnoukhov, A. (2012). Exploration of alpine orographic precipitation patterns with radar image processing and clustering techniques. Meteorological Applications, 19(4), 407-419. https://doi.org/10.1002/met.272

Wolvin, S., Strong, C., Rupper, S., & Steenburgh, W. J. (2024). Climatology of orographic precipitation gradients over High Mountain Asia derived from dynamical downscaling. Journal of Geophysical Research: Atmospheres, 129(20), e2024JD041010. https://doi.org/10.1029/2024JD041010

Gao, S., Zhang, J., Li, D., Jiang, H., & Fang, Z. N. (2021). Evaluation of multiradar multisensor and stage IV quantitative precipitation estimates during Hurricane Harvey.

*Natural Hazards Review, 22*(1), 04020057. https://doi.org/10.1061/(ASCE)NH.1527-6996.0000435